# Pili allow dominant marine cyanobacteria to avoid sinking and evade predation

Maria del Mar Aguilo-Ferretjans [1], Rafael Bosch [1,2], Richard J. Puxty [3], Mira Latva [3,4], Vinko Zadjelovic[3], Audam Chhun [3], Despoina Sousoni[3], Marco Polin [4], David J. Scanlan [3] & Joseph A. Christie-Oleza [1,2,3 ✉]

How oligotrophic marine cyanobacteria position themselves in the water column is currently unknown. The current paradigm is that these organisms avoid sinking due to their reduced size and passive drift within currents. Here, we show that one in four picocyanobacteria encode a type IV pilus which allows these organisms to increase drag and remain suspended at optimal positions in the water column, as well as evade predation by grazers. The evolution of this sophisticated floatation mechanism in these purely planktonic streamlined micro-organisms has important implications for our current understanding of microbial distribution in the oceans and predator–prey interactions which ultimately will need incorporating into future models of marine carbon flux dynamics.

[1] University of the Balearic Islands, Palma, Spain. [2] IMEDEA (CSIC-UIB), Esporles, Spain. [3] School of Life Sciences, University of Warwick, Coventry, UK. [4] Department of Physics, University of Warwick, Coventry, UK. ✉email: Joseph.Christie@uib.eu

A quarter of all primary production on Earth occurs in large nutrient deplete oceanic gyres[1]. Primary production in these large biomes is mainly driven by the dominant marine cyanobacteria i.e. *Prochlorococcus* and *Synechococcus*[2]. Gyres are permanently thermally stratified, where a lack of upward physical mixing poses a challenge for the microbial communities that inhabit them. How then do purely planktonic cyanobacterial cells in suspension combat the downward pull of gravity through the biological pump—i.e. drawing fixed carbon towards the ocean interior? Moreover, how do these highly specialised planktonic microbes place themselves in their 'preferred spot', such as the well-established vertical distribution of high and low light-adapted *Prochlorococcus* ecotypes[3–5]? Marine picocyanobacteria lack flagellar structures for swimming or gas vacuoles for floatation[6], and only a limited number of *Synechococcus* strains possess non-conventional mechanisms for motility[7,8]. Therefore, it has been assumed that these free-living microbes avoid sinking due to their lower density and reduced size[9], and bloom when they encounter their optimal environmental conditions while drifting randomly within marine currents.

To date, type IV pili are known to provide functions such as twitching motility, surface attachment, biofilm formation, pathogenicity, as well as conjugation, exogenous DNA acquisition and competence[10–12]. These extracellular appendages can be rapidly extended and retracted by polymerising and depolymerising cycles of the major pilin subunit e.g. PilA, requiring a defined transmembrane apparatus and the consumption of energy in the form of ATP[11,13]. Functional analysis of most type IV pili has focused on pathogenic microbes and their use of surfaces or substrates for pilus anchoring. However, analysis of pili from freshwater cyanobacteria has revealed these appendages can be used for twitching motility during phototaxis in *Synechocystis*[14] or exogenous DNA acquisition in both *Synechocystis* and *Synechococcus elongatus*[15,16]. This latter function requires an additional set of proteins for competence such as ComEA and ComEC. While the third of three PilA-like proteins (PilA3) encoded by *S. elongatus* is required for transformation[16], no known function has been attributed to PilA1 and PilA2 other than being dispensable for attachment and biofilm formation[17]. A mutant in *S. elongatus* that no longer produced PilB—the protein responsible for pilus elongation—abolished the production of pili appendages, made up of PilA1, and was reported to suppress planktonic growth of this strain[17]. The presence of pilus genes in purely planktonic marine microbes has previously been reported, but their role remains enigmatic[18].

Here, we show that almost a quarter of all marine picocyanobacteria encode a PilA1-like pilus. We show that this extracellular appendage produced in these purely planktonic organisms—which rarely encounter any kind of surface in their natural habitat—allows cells to increase drag and remain in an optimal position in the water column as well as avoid being preyed upon. This provides yet another biological function to these filamentous appendages and sheds light on the ecological role of type IV pili in marine ecosystems.

## Results and discussion

### Abundant production of a type IV pilus in *Synechococcus* sp. WH7803.
We first detected an abundant PilA protein (i.e. SynWH7803_1795) in the extracellular proteomes of the model marine cyanobacterium *Synechococcus* sp. WH7803, accounting for up to 25% of the exoproteome[19,20]. Transmission electron microscopy confirmed the existence of the macromolecular pili structures (Fig. 1a and Fig. S1). Unlike *Synechocystis* sp. PCC6803 that simultaneously produces thick and thin pili[14], this marine picocyanobacterium presented multiple pili of similar thickness

(diameter of ~6 nm), each ~10 μm in length. The amino acid sequence of PilA revealed a typical Sec-targeting signal peptide and a conserved GFTLxE motif at the N-terminus of the protein (Fig. 1b) that is known to be cleaved in the cytoplasmic membrane by PilD before the protein is translocated to the base of the pili for assembly[10]. After cleavage, the N-terminal of PilA can be post-translationally modified, e.g. methylated, to increase the hydrophobicity and stability of the pilin[10,21], although we were unable to detect this modified N-terminal tryptic peptide during proteomic analyses. In close proximity to *pilA* in the *Synechococcus* sp. WH7803 genome we found five other type IV-like pilin genes (Fig. 1c), all with the conserved GFTLxE motif (Fig. 1d).

Using the pilus apparatus from the freshwater cyanobacterium *S. elongatus* PCC 7942 as a reference[16] and the established architecture for type IV pilus machinery[11], we were able to find all components necessary for pilus assembly in *Synechococcus* sp. WH7803 (Fig. 1e). We speculate that the genetic cluster encoding the six pilin-like proteins (Fig. 1c and 1d) may provide three distinct pili functions. Based on homology with the annotated genes from *S. elongatus*[16] and conserved domains found using the CD-search tool in NCBI, we suggest the three pilin pairs: PilA1-PilE, PilA2-PilV and PilA3-PilW (Fig. 1c). Of these, shotgun proteomic analyses have only ever detected PilA1-PilE[19,20] implying these are responsible for the pili observed in Fig. 1a, although PilA2 was also detected in low abundance in cellular—but not extracellular—proteomic datasets[22]. Unlike in *S. elongatus*, where PilA1 and the contiguously-encoded pilin-like protein are almost identical, the amino acid sequence of PilA1 and PilE in *Synechococcus* sp. WH7803 are clearly distinguishable. Although in much lower abundance, PilE seems to be correlated with PilA1 in the exoproteomes of this cyanobacterium[19,20] and, therefore, it is possible that PilE and PilA1 form subunits of the same pilus apparatus.

### Pilus distribution amongst picocyanobacterial isolates and Single-cell Assembled Genomes (SAGs).
Genomic analysis of sequenced marine picocyanobacterial isolates downloaded from the Cyanorak database[23] revealed that 74% of sequenced *Synechococcus* ($n = 46$) and 33% of *Prochlorococcus* ($n = 43$) encoded *pilA1* (Fig. 2 and Supplementary Data 1). In *Synechococcus*, *pilA1* was prevalent in all clades (93%; $n = 28$) except for clades II and III where it was less abundant (44%; $n = 18$). Interestingly, all low light *Prochlorococcus* isolates from clades III and IV encoded *pilA1* ($n = 7$; Fig. 2). Most of these *pilA1*-containing strains also encoded a *pilE* homologue in close proximity (Fig. 2). Genes *pilA2* and *pilA3* were also abundantly found in *Synechococcus* (59 and 74%, respectively), although were much less prevalent in *Prochlorococcus* (12 and 9%, respectively). As expected, all strains that encode at least one of the *pilA* types also possessed the transmembrane pilus apparatus, whereas this apparatus was completely absent or partially lost in strains lacking *pilA* (Fig. 2). PilA3 is known to be involved in DNA uptake and competence in *S. elongatus*, requiring additional competence proteins to do so[16]. Marine picyanobacteria are not known for being naturally competent but, interestingly, all strains encoding PilA3 also contained the competence genes encoding ComEA and ComEC (Fig. 2). Further work is needed to investigate the conditions under which the PilA3-type pilus becomes active in these organisms and, therefore, when exogenous DNA might be taken up.

As well as encoded in their genomes, we further confirmed PilA1 was actively produced in other picocyanobacterial strains i.e. *Synechococcus* sp. BL107 (8.9% of its exoproteome[19]) and *Prochlorococcus* sp. MIT9313 (see below). Furthermore, pili similar to those observed in *Synechococcus* sp. WH7803 were actively assembled in *Synechococcus* sp. WH7805 (Fig. S1) despite the low homology between their *pilA1* genes and absence of a *pilE* homologue in the latter strain (Fig. 2).

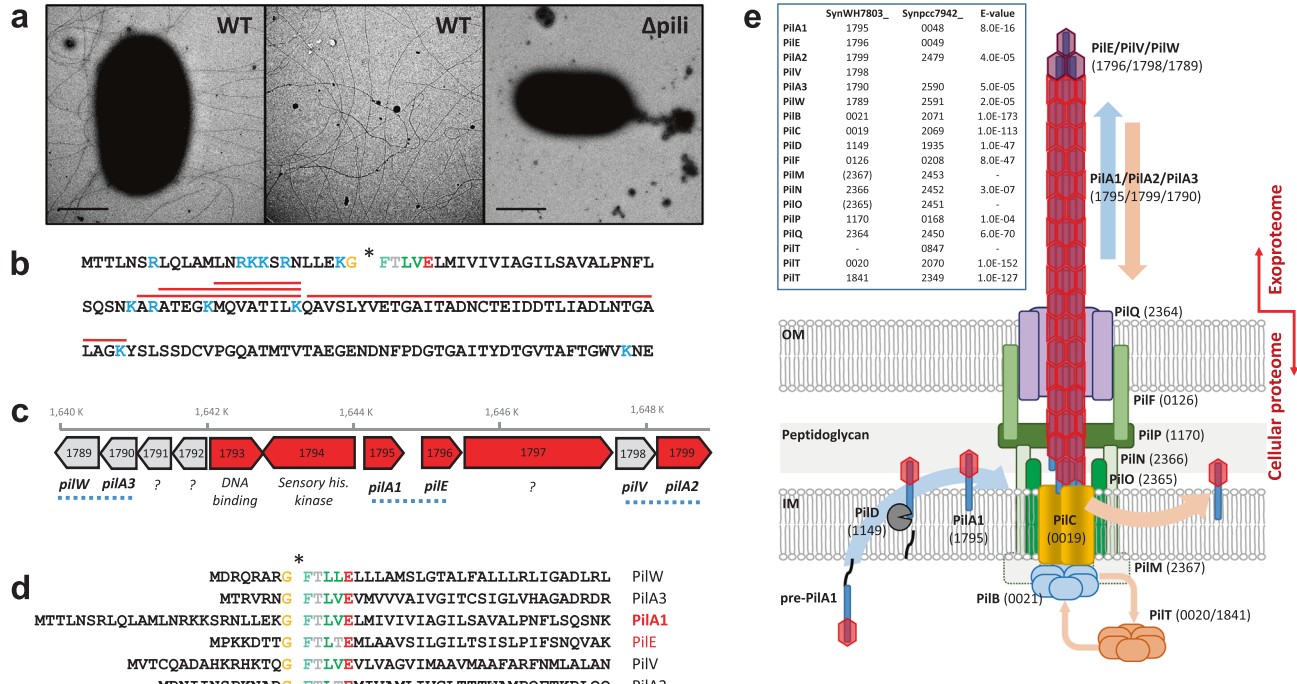

| | SynWH7803_ | Synpcc7942_ | E-value |
|---|---|---|---|
| PilA1 | 1795 | 0048 | 8.0E-16 |
| PilE | 1796 | 0049 | |
| PilA2 | 1799 | 2479 | 4.0E-05 |
| PilV | 1798 | | |
| PilA3 | 1790 | 2590 | 5.0E-05 |
| PilW | 1789 | 2591 | 2.0E-05 |
| PilB | 0021 | 2071 | 1.0E-173 |
| PilC | 0019 | 2069 | 1.0E-113 |
| PilD | 1149 | 1935 | 1.0E-47 |
| PilF | 0126 | 0208 | 8.0E-47 |
| PilM | (2367) | 2453 | - |
| PilN | 2366 | 2452 | 3.0E-07 |
| PilO | (2365) | 2451 | - |
| PilP | 1170 | 0168 | 1.0E-04 |
| PilQ | 2364 | 2450 | 6.0E-70 |
| PilT | - | 0847 | - |
| PilT | 0020 | 2070 | 1.0E-152 |
| PilT | 1841 | 2349 | 1.0E-127 |

**Fig. 1 Pilus in the marine cyanobacterium *Synechococcus* sp. WH7803. a** Transmission electron microscopy images of wild-type *Synechococcus* sp. WH7803 (WT) and pili mutant (Δpili) obtained from late-exponential liquid cultures incubated in ASW medium under optimal growth conditions. Imaging of three independent cultures in different occasions consistently showed long pili appendages only in the wild-type strain (Fig. S1). Middle panel image, obtained with the same magnification as other panels, is from an intercellular region between wild-type cells to improve the visualisation of the pili. Scale bar represents 1 μm. **b** The amino acid sequence of PilA1 (SynWH7803_1795). Trypsin hydrolytic sites are indicated in blue. Red lines highlight tryptic peptides detected by shotgun proteomics. The conserved GFTLxE motif is shown and the cleavage site is indicated with an asterisk. **c** Genomic context of *pilA1* in *Synechococcus* sp. WH7803. Numbers in each gene represent their ID number (SynWH7803_). In red are genes detected by proteomics. While PilA1 and PilE are abundantly detected in exoproteomes[20], PilA2 has only ever been detected in cellular proteomes of this strain[22]. Blue dotted lines indicate genes encoding possible structural pilin pairs, i.e. PilA1-PilE, PilA2-PilV and PilA3-PilW. Question marks indicate genes encoding proteins of unknown function. **d** The N-terminal amino acid sequence of PilA1 and five other pilin-like proteins, all with the highly conserved GFTLxE motif. **e** *Synechococcus* sp. WH7803 structural pilus proteins identified by homology with *S. elongatus* PCC 7942[16] and assembled in the inner (IM) and outer membrane (OM) as modelled by Craig et al[11]. *pilM* and *pilO* were not identified by homology but the SynWH7803_2367 and SynWH7803_2365 genes are suggested because they form a standard *pilMNOQ* operon as found in other species. While *S. elongatus* PCC 7942 encodes three *pilT*, only two were found in *Synechococcus* sp. WH7803, one being part of the characteristic *pilCTB* operon.

The GFTLxE motif is conserved in 87.5% of PilA1 sequences encoded by cultured marine picocyanobacteria (i.e. 42 of the 48 sequences; Supplementary Data 1). The remaining six PilA1 sequences possess a GFSLxE motif, five of which were in *Prochlorococcus* strains. Across the full length of the mature PilA1 protein, which on average is 140 amino acids long, only the first ~50 N-terminal amino acids starting from the conserved GFTLxE motif are well conserved amongst all sequences, a commonly observed feature in PilA-like proteins[11]. The C-terminus of the protein showed a remarkably high variability even between closely related strains. Despite this high variability, their predicted structures were still similar to those of known pili subunits (Fig. S2)[24,25]. During pili assembly, the helix encoded by the conserved N-terminus of PilA remains in the pilus core and only the variable C-terminus—that producing anti-parallel β-sheets—is exposed to the milieu[11]. We hypothesise that the hyper-variability of the exposed C-terminus is a strategy to escape phage attachment, it being a known pathway used by phage for host encounter and infection[26,27]. Similarly, flagella have a hyper-variable region, which has also been attributed to phage and immune system evasion[28].

The screening of 190 picocyanobacterial Single-cell Assembled Genomes (SAGs) obtained from surface seawater across the globe[29]—those with over 75% completeness—revealed the presence of *pilA1* and genes encoding components of the pilus apparatus in almost one in four marine picocyanobacteria (24% encoded *pilA1* and 21–25% the pilus apparatus; Table 1 and Supplementary Data 1). As expected by its abundance in the oceans, *Prochlorococcus* comprised almost 96% of all 190 SAGs (Table 1). The prevalence of the pilus was much higher amongst the low light *Prochlorococcus* SAGs from clades II/III (67–80%) than in those belonging to other ecotypes, and *Synechococcus* showed the *pilA1* prevalence observed in cultured isolates (~76%; Table 1).

**Global distribution and expression of picocyanobacterial *pilA1* in marine pelagic ecosystems**. The distribution and expression of *pilA1* in the surface ocean was determined by analysing its presence in the global marine TARA metagenome and metatranscriptome datasets. An HMM profile generated from the PilA1 sequences (cultured isolates and SAGs; Supplementary Data 1) was used to search the TARA datasets in the Ocean Gene Atlas portal[30] retrieving 903 and 837 individual hits from the metagenomes and metatranscriptomes, respectively (using a cut-off E-value $< 10^{-10}$). Sequences assigned to *Prochlorococcus* represented 85% in both datasets, whereas those assigned to *Synechococcus* represented 12% and 11% of the metagenomes and metatranscriptomes, respectively. BLAST analysis of these hits

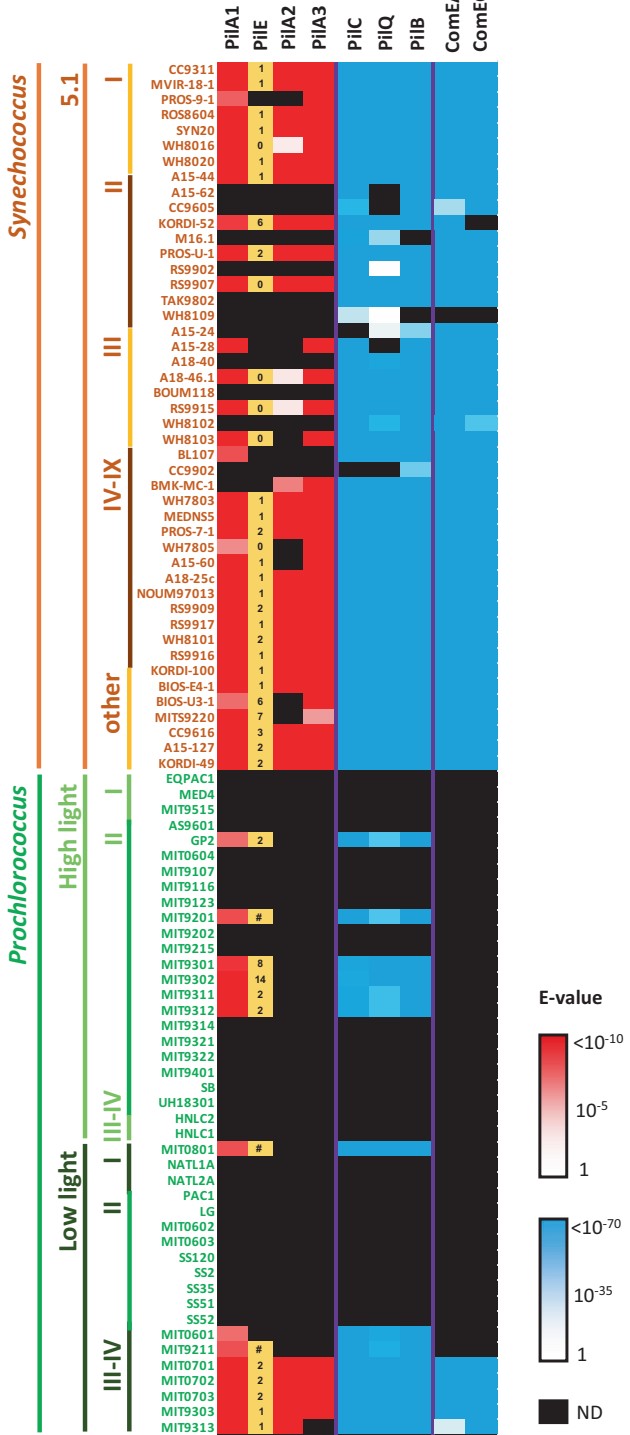

**Fig. 2 The presence of pilus-related proteins in cultured marine picocyanobacteria strains.** Pilus proteins from *Synechococcus* sp. WH7803 were used for the BLASTp search. Log10 E-value scales are shown (1 to <10⁻¹⁰, white to red; and 1 to <10⁻⁷⁰, white to blue). Black cells represent proteins that were not detected (ND). Numbers in the 'PilE' column indicate the genomic distance between *pilE* and *pilA1* homologues (e.g. 1 denotes *pilE* and *pilA1* are contiguous in the genome; 0 denotes the same gene gave homology to both *pilE* and *pilA1* due to the conserved N-terminal of the protein; # denotes both genes are separated by >20 genes). PilC, PilQ and PilB were used to determine the presence of the pilus transmembrane apparatus. ComEA and ComEC were selected to determine the presence of the additional machinery required for competence.

against PilA1, PilA2 and PilA3 sequences was used to confirm the specificity of our HMM profile, proving effective in discriminating against PilA2 and PilA3 sequences (each representing less than 1% of the hits).

The abundance and transcription of genes encoding PilA1 from *Prochlorococcus* and *Synechococcus* across all oceanic regions, marine biomes and water depths (Fig. 3) revealed a similar abundance of *pilA1* from *Prochlorococcus* in both metagenomic and metatranscriptomic datasets. In contrast, *pilA1* from *Synechococcus* was enriched in the metatranscriptomes, mainly driven by the increased expression in the North Atlantic Ocean and Mediterranean Sea as well as in coastal and westerlies, biomes where *Synechococcus* are known to thrive. As expected, a large reduction in the presence and expression of picocyanobacterial *pilA1* was noted in polar oceans, where both genera are not abundant. Furthermore, the presence and transcription of cyanobacterial *pilA1* decreased drastically in the aphotic mesopelagic layer, in accordance with picocyanobacteria naturally populating only euphotic layers of the ocean.

**PilA1-type pili increase drag and allow cells to remain planktonic.** The extensive distribution and expression of such an extracellular appendage begs the question why such a complex structure is so prevalent in such streamlined planktonic cyanobacteria. To assess this, and to assign a biological function to this extracellular structure in the context of the ecology of marine planktonic bacteria more generally, we abolished the production of PilA1 and PilE in *Synechococcus* sp. WH7803. As expected, the fully segregated pili mutant strain no longer produced the extracellular structure (Fig. 1a and Fig. S1). Most remarkably, we observed a clear loss in the strain's ability to remain suspended in its typical planktonic form (Fig. 4a). By tracking wild-type and pili mutant cells of *Synechococcus* sp. WH7803, we determined that the lack of pili produced an average cell sinking rate of 8.4 ± 0.4 mm/day, while the wild-type had an average uplifting drift of 0.8 ± 0.3 mm/day (Fig. 4b, Supplementary Movies 1–4 and Supplementary Data 2). While avoiding sedimentation, the pili did not appear to confer apparent twitching motility, with both mutant and wild-type strains producing 'pin-prick' colonies in sloppy agar plates as opposed to fuzzy colonies characteristic of motile[8] and twitching strains[14]. We further confirmed no apparent differences in cell size between the mutant and wild-type strains (2.05 ± 0.42 ×1.21 ± 0.11 μm and 2.02 ± 0.35 × 1.26 ± 0.13 μm, respectively, consistent with published cell size data[31]), and observed no cell aggregates when grown in shaken or non-shaken liquid cultures (Figs. S3 and S4).

We performed a comparative proteomics analysis between wild-type *Synechococcus* sp. WH7803 and the pili mutant to assess any additional effects of disrupting the PilA1-PilE pilus (Supplementary Data 3). Apart from the complete absence of the PilA1 and PilE proteins, only three other proteins were significantly downregulated in the cellular proteome of the pili mutant strain (Fig. S5): SynWH7803_0049 (12.9 fold) and SynWH7803_1797 (3.2 fold), both of unknown function and, most interestingly the pilus retraction protein PilT (SynWH7803_0020; 2.9-fold reduction). SynWH7803_1797 is located just downstream of PilA1-PilE and is predicted to encode a secreted protein that is usually found in low abundance in the exoproteome of *Synechococcus* sp. WH7803[20]. Indeed, it was also found downregulated in the exoproteome of the pili mutant (2.5×; Fig. S5). The exoproteomes also revealed a generalised shift of proteins more abundantly detected in the pili mutant. These were mainly low abundance cytoplasmic proteins that were barely detected in the wild-type strain (Fig. S5 and Supplementary Data 3). Most likely, the

**Table 1 PilA1 and pilus apparatus distribution among planktonic marine SAGs.**

|  |  | # SAGs[a] | Compl[b] | PilA1[c] | PilC[c] | PilQ[c] | PilB[c] |
|---|---|---|---|---|---|---|---|
| Total |  | 190 | 87% | 39 (24%) | 39 (24%) | 35 (21%) | 42 (25%) |
| *Prochlorococcus* | HL I | 23 | 89% | 4 (20%) | 5 (24%) | 3 (15%) | 5 (24%) |
|  | HL II | 98 | 88% | 12 (14%) | 12 (14%) | 12 (14%) | 13 (15%) |
|  | HL IV | 5 | 88% | 1 (23%) | 1 (23%) | 1 (23%) | 1 (23%) |
|  | LL I | 28 | 87% | 6 (25%) | 5 (21%) | 7 (29%) | 6 (25%) |
|  | LL II/III | 9 | 83% | 6 (80%) | 5 (67%) | 5 (67%) | 5 (67%) |
|  | Unclassified | 19 | 86% | 5 (31%) | 4 (24%) | 4 (24%) | 4 (24%) |
| *Synechococcus* |  | 8 | 82% | 5 (76%) | 7 (100%) | 3 (46%) | 8 (100%) |

[a]Total number of available SAGs with >75% genome completeness and for which the phylogeny had been assigned.
[b]SAGs average genome completeness.
[c]Number of SAGs encoding for the pilus proteins. The prevalence in each group is shown as the percentage of SAGs encoding each protein corrected by the average genome completeness.

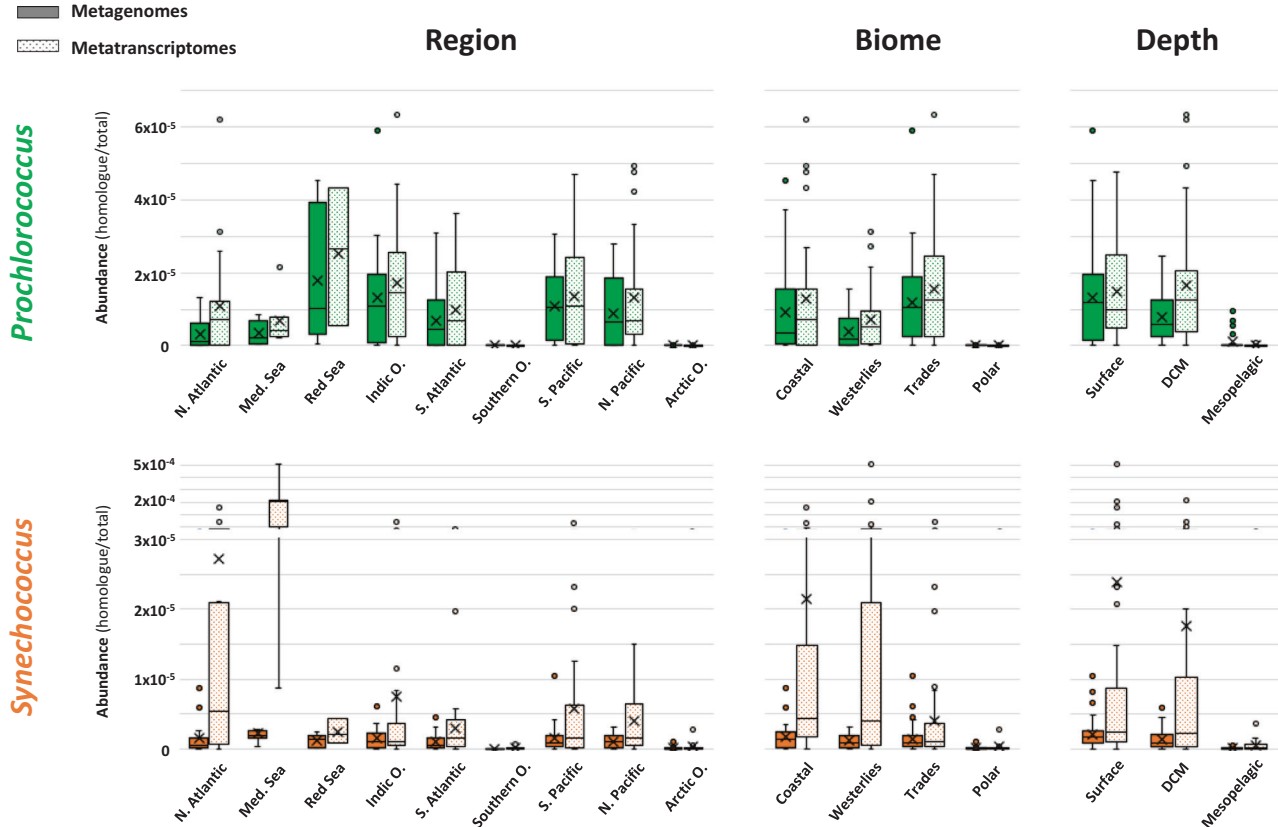

**Fig. 3 Distribution of PilA1 amongst marine ecosystems.** Whisker box plots showing *pilA1* gene abundance in *Prochlorococcus* (green) and *Synechococcus* (orange) in metagenomes (solid bars; $n_{mg} = 174$) and metatranscriptomes (dotted bars; $n_{mt} = 178$) generated from all filters (0.2–3 μm) obtained from 125 sampling stations of the TARA Oceans global marine survey. Abundance was calculated by dividing the sum of the abundances of *pilA1* homologs assigned to each genus by the sum of total gene abundance from all reads from the sample. Data are presented by oceanic region (North Atlantic Ocean, $n_{mg} = 23$, $n_{mt} = 21$; Mediterranean Sea, $n_{mg} = 12$, $n_{mt} = 7$; Red Sea, $n_{mg} = 6$, $n_{mt} = 3$; Indic Ocean, $n_{mg} = 27$, $n_{mt} = 22$; South Atlantic Ocean, $n_{mg} = 19$, $n_{mt} = 19$; Southern Ocean, $n_{mg} = 4$, $n_{mt} = 8$; South Pacific Ocean, $n_{mg} = 31$, $n_{mt} = 41$; North Pacific Ocean, $n_{mg} = 16$, $n_{mt} = 25$; Arctic Ocean, $n_{mg} = 36$, $n_{mt} = 32$), biome (Coastal, $n_{mg} = 35$, $n_{mt} = 33$; Westerlies, $n_{mg} = 37$, $n_{mt} = 28$; Trades, $n_{mg} = 62$, $n_{mt} = 103$; Polar, $n_{mg} = 40$, $n_{mt} = 40$) and water depth (surface, $n_{mg} = 62$, $n_{mt} = 78$; deep chlorophyll maxima (DCM), $n_{mg} = 43$, $n_{mt} = 39$; mesopelagic layers, $n_{mg} = 29$, $n_{mt} = 21$). Filters from polar regions were excluded from the depth abundance analysis. Whiskers indicate variability outside the upper and lower quartiles (boxes). Exclusive median (line), average (cross) and atypical values (circles) are also indicated. Source data are provided as a Source Data file.

absence of the abundant PilA1 protein from the exoproteome of the mutant strain caused an artifactual increase in the detection of lower abundance proteins by mass spectrometry and, despite efforts to normalise the data, there was an apparent upregulation of most proteins.

We had previously observed that *Prochlorococcus* sp. MIT9313 (a strain that encodes PilA1; Fig. 2) routinely switches between planktonic and sedimenting lifestyles when grown in different media (Pro99 and PCR-S11, respectively; Fig. 4c). Whilst *Prochlorococcus* remains genetically intractable[32], we compared the exoproteomes of this strain in both media. Commensurate with our findings in *Synechococcus* sp. WH7803, PilA1 was not detected in the exoproteomes of sedimenting *Prochlorococcus* sp. MIT9313 cultures (i.e. grown in PCR-S11 medium under shaking

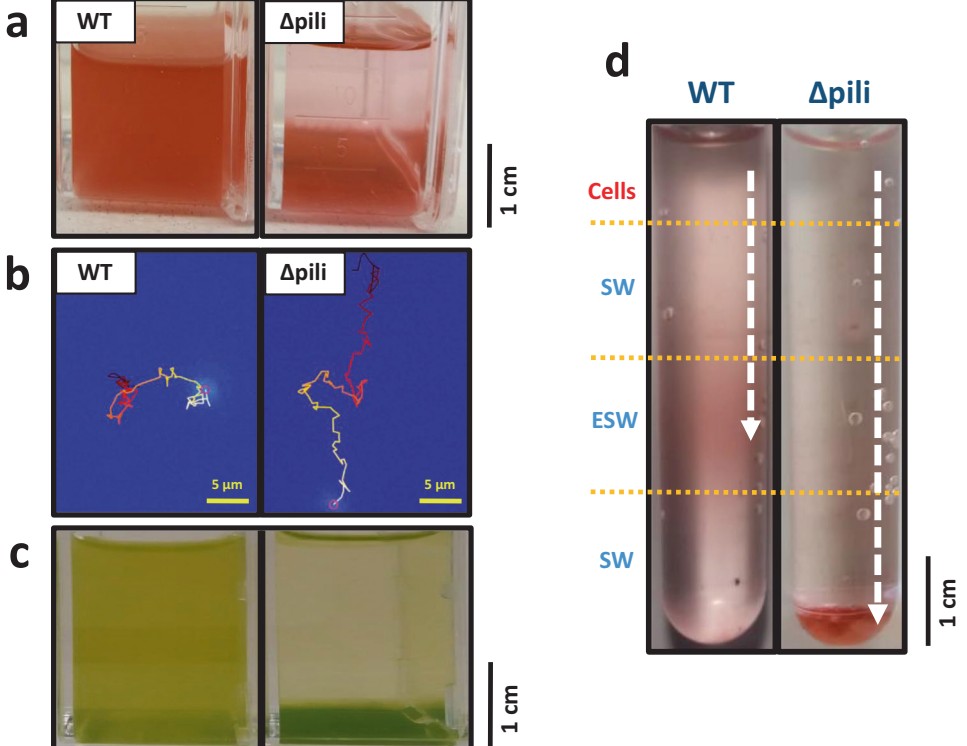

**Fig. 4 Pili avoid sinking in planktonic marine picocyanobacteria. a** Non-shaken cultures of wild-type (WT) and pili mutant (Δpili) *Synechococcus* sp. WH7803. The pili mutation and phenotype has proven very stable with the mutant strain unable to remain in a planktonic form when not shaken. The appearance of the mutant in the image is consistently achieved after 3–4 days of no shaking. **b** Example of a tracked suspended cell from a wild-type and pili mutant culture of *Synechococcus* sp. WH7803. The movement was tracked over 100 s and movement is indicated from dark red (start) to yellow (end). Full data of all tracked cells can be found in Supplementary Data 2 (*n* = 2614 WT and *n* = 439 pili mutant cells) and examples in Supplementary Movies 1–4. **c** Non-shaken *Prochlorococcus* sp. MIT9313 cultures grown in Pro99 medium (left) and PCR-S11 medium (right). The appearance of cells in the image is consistently achieved after 2–3 days of no shaking. PilA1 (PMT_0263) was only detected in the exoproteome of suspended cultures (left; Supplementary Data 4). **d** Nutrient step gradient column where wild-type and pili mutant cells of *Synechococcus* sp. WH7803 were placed at the top. Nutrient deplete (SW) and nutrient enriched layers (ESW) are indicated. Cells were harvested by centrifugation from late-exponential cultures grown under optimal conditions and resuspended in SW before placing them on top of the column. The arrow indicates the trajectory made by the cells, and where they accumulated after four days. Images represent one of three culture replicates. Figure S6 shows the quantification of the sinking and accumulation of cells in the different layers over time.

conditions for cells to remain planktonic), whereas it was present in all of the planktonic ones (i.e. PMT_0263 was detected in all cultures using Pro99 medium; Supplementary Data 4), suggesting *Prochlorococcus* can coordinate production of the pili in response to the distinct nutrient environments present in both media.

**PilA1 allows cells to retain an optimal position within the water column.** We further explored the effect of nutrient stress on the detection of PilA1 in *Synechococcus* sp. WH7803 exoproteomes, with nitrogen and metal depletion showing the strongest decline (i.e. 11.5 and 3.4-fold decrease in PilA1 production, respectively), while phosphorus depletion had no effect (Supplementary Data 5). Although in low abundance, PilA1 of *Synechococcus* sp. WH7803 was detected even under natural oligotrophic conditions (i.e. when incubated in natural seawater; representing 0.3% of the exoproteome) and showed a slight increase after adding environmentally relevant concentrations of nutrients (0.5% of the exoproteome in the presence of 8.8 μM N and 0.18 μM P; Supplementary Data 6). PilA1 detection became most obvious under higher nutrient conditions (60-fold increase in pili at 88 μM N and 1.8 μM P when compared to nutrient deplete seawater; Supplementary Data 6). Considering the biological significance of this

phenotype in the context of marine planktonic organisms— where the production of pili reduces sedimentation by increasing the viscous drag of the cell—the extension/retraction of pili would allow a cell to position itself at an optimal position in the water column, e.g. in patches of high nutrient availability. To further investigate this, we setup a nutrient step gradient. While the pili mutant, as expected, sank through the gradient independently of nutrient availability, the wild-type strain was able to position itself in the nutrient-replete layer where it remained, presumably, via the production of pili (Fig. 4d and Fig. S6). Sinking was confirmed neither to be caused by cell aggregation as previously reported[33] nor to a variation in cell shape and size (Figs. S3 and S4).

*Synechococcus* sp. WH7803 also retained pili over a 24 h period of darkness. Nevertheless, after three days under dark conditions— during which cultures are known to remain viable[34], the detection of pili dropped drastically (i.e. >20-fold drop in pili abundance in the exoproteome; Supplementary Data 7). Marine cyanobacteria occupying photic layers of the ocean will therefore keep their pili structures over normal diurnal light-dark cycles to maintain their position, but cells will cease to retain their position once they sink out of the euphotic zone, retracting their pili possibly as a strategy to recover energy while awaiting an uplift back to photic layers by upwelling currents.

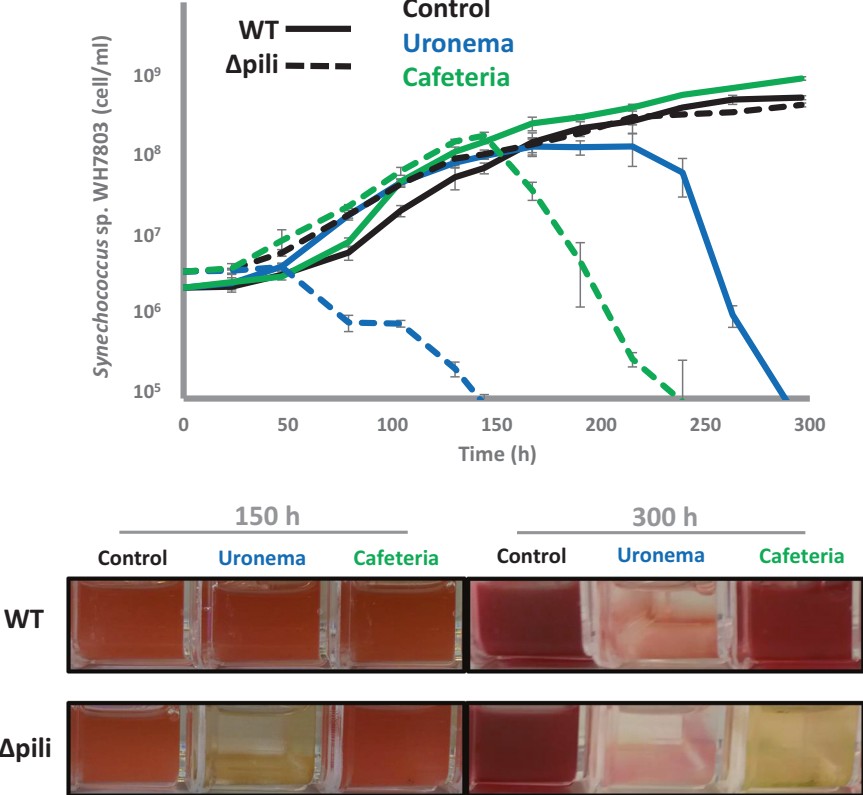

**Fig. 5 Pili confer resistance to grazing.** Growth curves (top panel) and culture images (bottom panel) of wild-type (WT; solid lines) and pili mutant (Δpili; dashed lines) cultures of *Synechococcus* sp. WH7803 incubated in the absence (control; black lines) and presence of two different grazers i.e. *Uronema* (blue lines) and *Cafeteria* (green lines). Cultures were subjected to constant shaking to keep the wild-type and pili mutant in planktonic form. Data are presented as mean values ± standard deviation from three culture replicates. Images represent one of three culture replicates.

**Pili prevent grazing.** Given the cell surface nature of this pilus and previous reports showing grazing inhibition by a giant cell surface protein (i.e. SwmB in *Synechococcus* sp. WH8102)[35], we assessed whether these cell appendages could mediate ecological interactions with other organisms. Thus, *Synechococcus* sp. WH7803 and the pili mutant grown in the presence of two bacterivorous protozoa i.e. the preying ciliate *Uronema* and suspension feeding flagellate *Cafeteria*, strikingly showed that whilst the wild-type strain was able to completely evade grazing by *Cafeteria* and largely delay culture depletion by *Uronema*, the pili mutant was efficiently grazed by both protists (Fig. 5). Presumably, the long pili appendages interfere with the way bacterivorous protozoa access their prey and, hence, this reduces their susceptibility to grazing.

Together, our data suggest a novel and sophisticated mechanism that enables oligotrophic marine picocyanobacteria to stay buoyant and reduce cell death due to grazing through the use of pili, adding a new biological function to these extracellular appendages. This pilus, widely distributed and expressed amongst dominant marine oligotrophic cyanobacteria, allow these purely planktonic organisms to increase cell drag and, consequently, maintain an optimal position in the water column. This mechanism responds to discrete stimuli e.g. nutrients, which may vary depending on the ecological adaptation and preferred niche of each individual organism. Therefore, as opposed to flagellated bacteria that show positive chemotaxis towards nutrient hotspots in the oceans[36], picocyanobacteria may apply a more passive strategy which consists of elongating their pili when they encounter preferable conditions to remain in an optimal position while drifting in a water body. These long appendages also interfere with the access of bacterivorous protozoa to their prey allowing pili-producing cells to evade grazing. Further research should define: (i) the biophysical differences between pili that allow attachment and floatation, (ii) the resources required for this floatation system and advantages over other mechanisms such as flagella and (iii) the ecological trade-offs of having such extracellular appendages. Thus, besides being beneficial, pili could also be a handicap to those cells that produce them as, being phage binding sites[26,27], they will increase the chance of interacting with phage and hence their susceptibility to phage infection.

The biological carbon pump and microbial loop pose important challenges for streamlined marine cyanobacteria, which have moved away from canonical flagellar motility and have evolved this more passive mechanism for floatation that may require less resources and no additional convoluted tactic systems. This discovery changes our ecological perception of this dominant marine bacterial group, and will have important consequences for our future understanding of predator-prey and carbon flux dynamics in the oceans.

## Methods
**Culture conditions.** *Synechococcus* sp. WH7803 was grown in ASW medium[37] and autoclaved oligotrophic seawater[22]. Experiments were performed using 20 ml cultures contained in 25 cm² rectangular cell culture flasks (Falcon) with vented caps. Cultures were incubated under optimal growth conditions i.e. at 22 °C at a light intensity of 10 μmol photons m$^{-2}$ s$^{-1}$ with shaking (140 r.p.m.), unless otherwise stated in the text. To study the influence of nutrients on pili production, media was prepared by (i) not adding different nutrient sources into ASW media, i.e. nitrogen, phosphorus and trace metals, or (ii) diluting ASW in oligotrophic seawater (i.e. 1:1000, 1:100 and 1:10). The low light-adapted ecotype *Prochlorococcus* sp. MIT9313 was grown in

40 ml Pro99 medium and PCR-S11 medium with no additional vitamins[38]. Different light intensities (i.e. 4 and 15 µmol photons m$^{-2}$ s$^{-1}$) and temperatures (i.e. 14 and 22 °C) were tested. Cyanobacterial culture growth was routinely monitored by flow cytometry (BD Fortessa) operated by BD FACSDiva™ software.

Grazing experiments were performed using ASW-washed *Uronema marinum* (isolated from Qingdao Bay) and *Cafeteria roenbergensis* (strain CCAP 1900/1) cells. Briefly, 10 ml culture was subjected to centrifugation at 4000 × *g* for 15 min and the pellet resuspended in 10 ml ASW. One millilitre of washed grazers was used to inoculate 20 ml cultures of wild-type and pili mutant of *Synechococcus* sp. WH7803 at an initial concentration of 3–4 × 10$^6$ cells ml$^{-1}$. Triplicates cultures were incubated under optimal conditions, including shaking to avoid pili mutant sedimentation (see above).

**Pilus knockout mutant in *Synechococcus* sp. WH7803.** Genes SynWH7803_1795 and SynWH7803_1796 (*pilA1* and *pilE*, respectively) were replaced by a gentamicin cassette via a double recombination event to generate the *Synechococcus* sp. WH7803 pili mutant. Two flanking regions of 1000 bp from the genome of *Synechococcus* sp. WH7803 and the gentamicin cassette from pBBR-MCS[39] were amplified by PCR using primers indicated in Supplementary Table 1, and inserted into vector pK18mobsacB[40] using the Gibson assembly method following the manufacturer's instructions (New England Biolabs). The detailed protocol to generate mutants in *Synechococcus* sp. WH7803 is given in the Supplementary Note 1 and Fig. S7. All three transconjugant colonies that were picked had the same sinking phenotype, had doubly recombined (as checked by sequencing the overlapping regions) and were fully segregated. One mutant was subsequently selected to make axenic by eliminating the 'helper' strain and used for further experimentation. The pilus mutation is stable and has retained its sinking phenotype over time.

**Microscopy**

*Transmission Electron Microscopy.* 1.5 ml of optimally-grown *Synechococcus* sp. WH7803 and pilus mutant cultures were concentrated by centrifugation at 3000 × *g* for 3 min at room temperature, resuspended in 50 µl of fresh media and incubated under optimal conditions for 30 min. Then, 3 µl were delicately transferred onto a glow-discharge formvar/carbon coated grid and left 2–3 min for cells to attach. After blotting the excess media, negative staining was achieved by applying a drop of 2% uranyl acetate to the grid for 1 min. The excess stain was blotted off and left to air dry before imaging using a JEOL 2011 TEM with Gatan Ultrascan.

*Fluorescence microscopy.* Ten microlitre of optimally-grown *Synechococcus* sp. WH7803 and pilus mutant cultures (incubated under shaking and non-shaking conditions) were transferred to a glass microscope slide and immediately imaged with a confocal microscope (Leica TCS SPE, Leica Microsystems). Images were obtained using cyanobacterial autofluorescence and processed using the software Leica application suite (Leica Microsystems).

Cells (length and width) and pili (length and diameter) were measured using ImageJ (version 1.53a).

**Tracking and imaging of sinking cells.** The movement of wild-type *Synechococcus* sp. WH7803 and pilus mutant through a nutrient step gradient was performed by placing washed cells in oligotrophic SW on top of a water column. Nutrient deplete (oligotrophic SW) and nutrient amended layers (ASW) were placed below the top cell-containing layer as indicated. Density separation between layers was achieved by an increasing sucrose concentration (i.e. 2.5% w/v per layer).

Sedimentation tracking and velocity measurements were conducted using a setup as previously described[41]. Briefly, a sample chamber was prepared by gluing a square glass capillary (inner dimensions 1.00 × 1.00 mm, length 50 mm; CM Scientific, UK) onto a glass slide using an optical glue (#81; Norland, USA). Tubings (Masterflex Transfer Tubing, Tygon® ND-100-80 Microbore, 0.020" ID × 0.060" OD; inner dimension 0.51 mm; Cole-Parmer, USA) were attached to both ends of the sample chamber using blunt dispensing nozzle tips (Polypro Hubs; Adhesive Dispensing, UK), one-way stopcock valves (WZ-30600-00; Cole-Parmer, USA), and Luer connectors. *Synechococcus* was pulled into the capillary through the inlet by manual suction using a 2 ml syringe connected to the outlet of the capillary system. After introducing the sample, the microfluidic system was isolated by closing the stopcock valves. Samples were left to settle for 30–60 min before recording. The slide-capillary system was held vertically by an adjustable translation stage (PT1B/M; Thorlabs, USA), and placed between a white LED ring light source and a continuously focusable objective (InfiniProbe TS-160; Infinity Photo-Optical Company, USA) for dark-field imaging. Images were acquired with magnification set at ×16, using a CMOS FLIR Grasshopper3 (GS3-U3-23S6M-C; Point Grey Research Inc., Canada) operated with FlyCapture2 (FLIR Systems UK). Recordings were done at 1 fps for either 1, 2 or 5 min. The system was calibrated with a resolution target (R2L2S1P Positive NBS 1963A; Thorlabs, USA). Sedimentation velocities were calculated with custom-made codes in MATLAB 2019a, based on particle tracking code from Crocker and Grier[42]. Briefly, features corresponding to individual cells were selected based on shape and image intensity, and the distribution of frame-to-frame displacements was calculated from all the tracks lasting at least 20 s. All the experiments were checked for outliers.

**Proteome preparation and shotgun analysis.** Twenty millilitre cell cultures were subjected to centrifugation at 4000 × *g* for 15 min at 4 °C. Cell pellets were used for cell proteome analyses whereas the supernatants, which were further filtered to remove any remaining cells (0.22 µm), were used for exoproteomic analyses. Exoproteomes were concentrated using a trichloroacetic acid precipitation protocol as previously described[43]. Cell pellets and exoproteome precipitates were, respectively, resuspended in 100 µl and 50 µl of 1× LDS buffer (ThermoFisher) containing 1% beta-mercaptoethanol followed by three successive rounds of 5-min incubations at 95 °C and 5-min water-bath sonications with short vortex steps in between. Twenty microlitre were loaded on an SDS-PAGE precast 10% Tris-Bis NuPAGE gel (Invitrogen), using MOPS solution (Invitrogen) as the running buffer, allowing a protein migration for 5 min at 200 V. The resulting gel was stained using SimplyBlue Safe-Stain (Invitrogen) to visualize and excise the gel bands containing the proteomes. Polyacrylamide gel bands were destained and standard in-gel reduction and alkylation were performed using dithiothreitol and iodoacetamide, respectively, after which proteins were in-gel digested overnight with 2.5 ng µL$^{-1}$ trypsin[43]. The resulting peptide mixture was extracted by sonication of the gel slices in a solution of 5% formic acid in 25% acetonitrile, and finally concentrated at 40 °C in a speed-vac[44]. For mass spectrometric analyses, peptides were resuspended in a solution containing 0.05% trifluoroacetic acid and 2.5% acetonitrile, and further filtered using a 0.22 µm cellulose acetate spin column (Sigma–Aldrich)[44]. Tryptic peptides were analysed with an Orbitrap Fusion mass spectrometer (Thermo Scientific) coupled to an Ultimate 3000 RSLCnano system (Dionex), using conditions described in Christie-Oleza et al.[45]. Mass spectra were identified using *Synechococcus* sp. WH7803 and *Prochlorococcus* sp. MIT9313 UniProt databases (downloaded on 09/11/2017) and quantified using LFQ default parameters in MaxQuant v1.6.10.43[46] including the function of matching between runs. Comparative proteomic analyses were performed using MS intensity values in Perseus v1.6.2.2[47]. Briefly, proteins were filtered by removing decoy and contaminants and were considered valid when present in at least two replicates within one condition. The relative abundance of each protein was normalized to protein size and then to total sample signal.

**In silico analysis of pilus proteins.** Major pilin proteins and pilus machinery were searched in *Synechococcus* sp. WH7803 using reference proteins from *S. elongatus* PCC 7942[16] and proposed modelled structure[11]. Genomes from cultured picocyanobacteria (*Synechococcus*, n = 46, and *Prochlorococcus*, n = 43) were downloaded from Cyanorak v2.1[23] and re-annotated using PROKKA vs. 1.7[48]. Genomes were then screened for the presence of PilA-like proteins, structural proteins PilQ, PilB and PilC, and competence proteins ComEA and ComEC, via a local BLAST server using the amino acid sequences from *Synechococcus* sp. WH7803.

The 190 picocyanobacterial SAGs from Berube et al.[29], all with over 75% estimated genome completeness, were downloaded and annotated in-house using PROKKA vs. 1.7. Annotated SAGs were screened for pilus associated proteins using those from *Synechococcus* sp. WH7803 (Supplementary Data 1), with further verification using the Conserved Domain search tool from NCBI and manual curation in order to eliminate redundant matches within each SAG and select PilA1-like proteins which had a PilE-like protein encoded in close proximity in the genome.

The PilA1 sequences obtained from the cultured isolates and SAGs were used to generate an HMM profile in Unipro UGENE vs. 33[49] implemented with the hmmbuild programme from HMM3[50] using the default parameters. The resulting HMM profile was used to search the TARA oceans metagenomes and metatranscriptomes via the functions offered in the Ocean Gene Atlas portal[30].

The modelled 3D protein structure of PilA-like proteins was performed with mature sequences using the I-TASSER server's default settings[51].

**Reporting summary.** Further information on research design is available in the Nature Research Reporting Summary linked to this article.

## Data availability

All detailed methods and data are available as supplementary information and data. The RAW mass spectrometry proteomics data have been deposited in the ProteomeXchange Consortium via the PRIDE partner repository with the dataset identifiers: PXD018394, PXD018395, PXD018396, PXD018524 and PXD019315. Cultured marine picocyanobacterial genomes were downloaded from the Cyanorak database v2.1 [http://application.sb-roscoff.fr/cyanorak] (ref. [23]). Picocyanobacterial SAGs were downloaded from ref. [29] (https://figshare.com/collections/Single_cell_genomes_of_i_Prochlorococcus_i_i_Synechococcus_i_and_sympatric_microbes_from_diverse_marine_environments/4037048; doi: 10.6084/m9.figshare.c.4037048.v1). TARA metagenomes and metatranscriptomes were analysed using the Ocean Gene Atlas portal [http://tara-oceans.mio.osupytheas.fr/ocean-gene-atlas] (ref. [30]). Source data are provided with this paper.

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

## Acknowledgements

We thank Dr Bakker and the Advanced Bioimaging Platform as well as Dr Hernandez-Fernaud and the WPH Proteomic Facility, at the University of Warwick, for respective support in imaging and proteomics. We also thank the technical assistance by Dr Ramis at the Cellomics Unit (IUNICS, SCT) and Dr Hierro (SCT) of the University of the Balearic Islands. We thank Dr Guillonneau for providing the protist cultures and WISB centre (grant ref. BB/M017982/1) for access to equipment. This work was supported by NERC Fellowship NE/K009044/1 and Ramón y Cajal contract RYC-2017-22452.

## Author contributions

J.C.-O. conceived the study. M.A.-F. generated the knockout mutant. J.C.-O. and M.A.-F. performed the sinking experiments. V.Z., A.C. and D.S. performed the proteomic analyses. J.C.-O., R.B. and R.J.P. performed the bioinformatics analyses. D.J.S. aided the *Prochlorococcus* work. M.L. and M.P. carried out the tracking and imaging of sinking cells. J.C.-O. and M.A.-F. wrote the manuscript with large input from D.J.S. and R.J.P.

## Competing interests

The authors declare no competing interests.

**Additional information**

