## [Peer Review File · Nature Communications]

Reviewers' Comments:

Reviewer #1:

Remarks to the Author:

This manuscript presents interesting data on the distribution of genes encoding components of the type IV pilus in marine cyanobacteria. This was a so far weakly studied research subject and requires definitely more attention. The authors show that many pilA-like genes are widely distributed among *Prochlorococcus* and *Synechococcus* species based on metagenomic and metatranscriptomic data and that also other components of a typical type IV pilus machinery are encoded by many marine cyanobacteria. Then they have used a genetically tractable marine organism, *Synechococcus* sp. WH7803, to perform functional studies and demonstrate that a generated mutant lacking pili shows a phenotype related to the performance of these planktonic strain in a water column. The main conclusion from these data is that type IV pili help marine cyanobacterial strains to stay suspended in a water column and to avoid sinking. Further, they have evidence that pili are important to evade predation by grazers.

I have reviewed a previous version of this article and surely the authors improved it substantially in accordance with the previous comments of the reviewers. However, the manuscript still suffers from insufficient knowledge of the current literature on T4P and understanding of the key biological concepts of the function of this nanomachine. Therefore, several conclusions made by the authors cannot be drawn based on the presented data which do not rule out alternative explanations. In addition, the study still lacks important control experiments and proper statistical analysis.

Major comments:

1. line 55 and elsewhere (e.g. lines 92-95): In my opinion the authors have a conceptual misunderstanding of the pilus machinery and its function. It seems that the major pilin is PilA1 and all the other pili-like proteins are minor pilins. There is most probably nothing like a PilA1-like pilus or a PilA1-PilE, PilA2-PilV or PilA3-PilW pilus. As the authors detected only PilA1 and PilE in the exoproteome, it is likely that these pili subunits from the filament seen in EM. For the other so called minor pilins it is known that they can have various functions, such as initial assembly of the pilus, forming pilus tip, signaling function or DNA binding. Whether these other pilins are really part of a distinct filament or part of the PilA1 filament or have other functions is complete speculation. Usually, the minor pilins do not form a whole filament, instead they fulfill functions related to the function of the major pilus. Therefore, without further experimental data nothing can be speculated about these genes.

2. A major problem is the construction of the mutant. They found the downstream protein SynWH7803_1797 was also decreased in the proteome of the pilA1-pilE mutant. It is not clear, whether this is a polar effect of the mutation within one operon.

Generally, what is needed to be able to make clear conclusions about the mutant study is:

- i) To show transcriptomics data to give a hint which genes are cotranscribed in this region of the genome and where is a possible promoter.
- ii) Complementation of the mutant with pilA1 and pilE to discriminate between the role of both pilin genes and to exclude polar effects on downstream or maybe also upstream genes.

In general, I find it dangerous in bacterial genetics to work with only one clone without successful complementation of the phenotype.

3. Fig. 1A and Fig. S1: It seems that the mutant is smaller than the WT. Is this statistically relevant and do you see other obvious phenotypes in e.g. pigmentation or growth? It would be really beneficial to have a more comprehensive characterization of the mutant in order to exclude that the detected effects (related to sinking and grazing) are really due to the lack of pili and not because of size difference or different accumulation of carbohydrates. It is known from other cyanobacteria that the inactivation of PilA1 might have a pleiotrophic phenotype. Interestingly, there are only very small changes in the proteome of the pili mutant which does not hint to a major effect of the pilA1-pilE deletion on other cellular functions. However, changes can occur also on the level of protein activity controlled e.g. by second messenger molecules.

I have a further problem with the EM figures. As I have pointed out already in my last review on this

paper, this marine *Synechococcus* strain is known to be around 1 – 1.5 micrometer in size, here it seems that it is at least 4 micrometer which would be the size of *Synechococcus elongatus*. Whereas, the mutant has roughly the correct size of what you would expect for this organism. Do you have any explanation?

4. Line 251: Are there any transcriptomic data which would show that *Prochlorococcus* can really regulate the production of pili in response to different media? Maybe it is sedimentation (the surface contact) which controls pilus production.

Line 268: again, "... where it remained via the production of pili." This is an over interpretation of the data. You did not show that this is due to the production of pili. This could be also a secondary effect of lack of pili. It is e.g. shown that mechanosensing via type IV pili induces a signal transduction chain via second messengers which leads to activation of different cellular responses. Further, large amounts of PilA1 can be also detected in the expoproteome when pilus retraction is disturbed (e.g. pilT mutant strain). That means that you still can detect PilA, even if the pilus might be functionally inactive. In general, the authors should be more careful about what they have really measured and what can be concluded from these measurements.

5. What is the diameter of the pili? Does it fit to a typical type IV pilus? This should be measurable from the EM pictures.

6. Figure 1E: Several aspects in this figure are not correct: PilB and PilT are hexamers. PilA are localized in the membrane before they are incorporated into the pilus structure. They are not cleaved in the cytoplasm. The picture is here completely misleading. PilC is a single subunit complex. Didn't they detect PilM? What is with PilT1, they show PilT2 and PilT3. On which base a PilT3 protein was assigned?

7. In my opinion, the authors should try to quantify the data shown in Figure 4.

Minor comments:

Line 42: *Synechococcus elongatus* also performs phototaxis using type IV pili and most probably PilA1 (Yang et al., 2018).

Line 44: I think it is not quite correct to say that PilA3 performs acquisition of DNA, it was only shown that it is needed for transformation, but the exact function is unknown to my knowledge.

Line 106 and elsewhere: the authors mix up the terms pilin and pilus. Here, you most probably mean the pilin. You can't say from genomic data whether a pilus is present. You can only say whether the genes encoding pilus subunits are present in genomes.

Line 117 (see also major comment 1): You did not show that there is something like a PilA3-type pilus. You also did not show that this minor pilin can bind DNA. To my knowledge, it is also not possible to predict functions of these minor pilins from sequence. However, the authors did this in a large extent. Anyway, it seems the similarity between e.g. PilA3 from *Synechococcus elongatus* and the 1790 protein sequence from WH7803 is so low that I wonder on which base you think they have similar functions. The same is also true for the assignments of the other pilin.

255: better PilA1 accumulation, you do not measure pili production

260: you do not detect pili, this you can do e.g. by EM

296: There is nothing like a PilA1-type pilus (see comments above)

299: I think there is no evidence, that this regulation is tight and it is unclear what kind of regulation is meant here. There are no data on that.

302: Can you exclude that picocyanobacterial which harbor the genes for T4P are nonmotile?

313: Can you provide citation which prove that the genomes marine strains do not encode components of chemotaxis systems?

Fig. S3B: what is platocyanin? Do you mean plastocyanin?

Fig. S2: Please, check the legend, I see only models for PilA1, not for PilA2, PilA3 and PilA.

Reviewer #2:

Remarks to the Author:

The authors present genetic evidence of the encoding of a type IV pilus by the abundant marine

picocyanobacteria *Synechococcus* and *Prochlorococcus*. They applied culture experiments using *Synechococcus* isolates of wild-type (WT) and mutant strains lacking the pili that suggested that these proteins may increase a cell's viscous drag and aid in avoiding predation in the natural environment.

This is a comprehensive and very nicely presented study, and I found the genetic work particularly enticing and elegant, complemented by genomic analyses of known marine cyanobacterial isolates and single cell genomes. The expression and distribution of *pilA1* in the TARA metagenome and metatranscriptomes provided a better understanding of the distribution of these pili proteins in the natural environment, suggesting an ecological role. The methodology to produce *Synechococcus* mutants lacking pili proteins developed in this study should prove helpful for further studies of the physiological and biophysical mechanisms underlying the production and role of these pili in the natural environment.

I mainly have issue with the ecological interpretations and conclusions from the culture work, which in my view are too far reaching and not convincingly supported by biophysics nor the results of the experiments comparing WT vs mutant strains. I detail these below:

1. The difference in daily sinking rate of the pili mutant compared to the slight uplift is very small (Line 207 and following). While this may make difference in a test tube, in the ocean, both would be considered non-sinking. Also, in Fig. 4, the sinking rate of 8.4 mm/day determined from the tracking experiments would not explain the accumulation of the majority of pili mutant cells on the bottom of the tube after three days. Alternatively, the pili mutants might be physiologically different and show an increase in sinking rate. For example, the lack of pili could allow for more cell-to-cell contact and aggregate formation, thus enhancing sinking rates.

2. Similarly, the accumulation of WT cells in the nutrient rich layer of the tube in Fig. 4D could simply be due to an accumulation of cells that exhibit an increased growth rate. And it seems like some cells still sink beyond the ESW layer. Ultimately, the authors need to develop a biophysical explanation, e.g., using the Navier-Stokes equation, on the mechanism how pili could increase drag in a regime where $Re \ll 1$.

3. Also, when checking pilus distribution among picocyanobacteria, in clades II and II the gene was less abundant. However, these clades are the most abundant in the Sargasso Sea for example (Ahlgren and Røcap 2012), so one wonders why those clades do not have this gene if it is deemed so important for maintaining suspension in the water column. This appears inconsistent with the arguments presented regarding the importance of the pili proteins in these abundant picocyanobacteria.

4. One of the abundant strains of *Synechococcus* (WH8102) has been found to produce transparent expolymeric particles (TEP) in culture, increasing stickiness of the cells and leading to formation of cell aggregates (Cruz et al., 2019). This could interfere with the postulated effect of the presence of the pili in decreasing sinking rate, and should be discussed.

5. Regarding the grazing experiments: previous studies on the role of surface proteins have already shown inhibition of grazing on *Synechococcus*. Strom et al., 2012 found that mutant cells of Syn WH8102 lacking the SwmB protein were more susceptible to being grazed compared to the WT, which could be parallel to what was found herein with pili-lacking mutants, and needs to be discussed.

Minor comments:

1. The methods are clearly written and referenced by supplemental material, detailed enough to allow for reproducibility, especially for production of *Synechococcus* mutants. However, to make the nutrient step gradient experiments more reproducible (lines 359-362), the authors could present more details

on the addition of the sucrose. Was it added just to create the different density layers? This needs to be clarified.

2. Line 46: the wild-type of *S. elongatus* was already a biofilm former; the referenced study showed that removing the pili didn't stop biofilm formation, just occurred more slowly - the cells were not planktonic even in wild type.

3. Line 68: modify to "N-terminal"

4. Line 332: what is the meaning of "CCAP 1900/1 cells"?

References:

Ahlgren, N. A., and G. Rocap. 2012. Diversity and distribution of marine *Synechococcus*: Multiple gene phylogenies for consensus classification and development of qPCR assays for sensitive measurement of clades in the ocean. *Front. Microbiology* 3: 1–24. doi:10.3389/fmicb.2012.00213

Cruz, B.N. and S. Neuer. 2019. Heterotrophic bacteria enhance the aggregation of the marine picocyanobacteria *Prochlorococcus* and *Synechococcus*. *Frontiers in Microbiology*, doi: 10.3389/fmicb.2019.01864

Strom, S. L., Brahmsha, B., Fredrickson, K. A., Apple, J. K., and A. G. Rodríguez. 2011. A giant cell surface protein in *Synechococcus* WH8102 inhibits feeding by a dinoflagellate predator. *Environ. Microbiol.* 14: 807–816. doi:10.1111/j.1462-2920.2011.02640.x

Reviewer #3:

Remarks to the Author:

Overall comments

The manuscript entitled "Pili allow dominant marine cyanobacteria to avoid sinking and evade predation" by Aguilo-Ftrretjans et al deal with the function of type 4 pili. They found that type 4 pili encoding genes are present in about one quarter of picocyanobacteria. The authors claim that pilus filaments confer two biological important functions to marine picocyanobacteria, one is to allow cells to float at nutrient rich region in the water column, and another is to avoid predation by grazers. These findings are novel, and I believe they are of great interest to cyanobacterial field and maybe to a broader field of ecology.

The manuscript is clearly written, and the important experimental details are provided. However, some statements in the text are not well supported.

My major concern is the experimental evidences that support these conclusions are based only on the model organism *Synechococcus* sp. WH7803. To make a general conclusion for marine picocyanobacteria, the author should perform the same sinking experiment and grazing experiment with 2-3 more pilated strains.

Major comments:

Figure 1:

1. Why all the pili mutants (also from Figure S1) seem to have a smaller cell size?

2. The drawing for pili could be misleading. PilD is a membrane protein, not cytoplasmic; PilB and PilT are hexamer, not octamer; PilM is missing and there is no mention of this protein at all throughout the manuscript, is it because there is no PilM homolog in WH7803? How conserved is this protein in bacterial type 4 pili?

Figure 4:

1. Could the authors perform the cell sinking experiment for a different piliated picocyanobacterial strain if they can find one? So that you can consider it a general mechanism but not just exist in WH7803.
2. Based on the movies 1-2, mutant cells seem a little bit longer than the WT (but the TEM images show a smaller cell size of the mutant). Since the cell size/shape may affect the sinking rate, I think it is necessary to compare the mutant cell size to the wild type, to make sure that pili mutant does not affect cell size.
3. I did not find in the material and method that how the cells are labelled for the tracking experiment in panel B and movies 1-4.

Figure 5:

1. Likewise, it would be more convincing if the authors can show that the same result is obtained with another piliated picocyanobacterial strain.

Line 63: how do you get the number of 10 μ m length for the pili, it is hard to see the end of a pili from the TEM images, they seem all twisted around each other, which make it difficult to trace.

Line 90-91: you stated that you find all components necessary for pilus assembly, but I don't see the structural protein pilM, which I think is important for the pili function.

Line 210-212: the conclusion of "pili did not appear to confer motility" is not appropriate here purely based on the comparison of the colony shape with the *Synechococcus* sp. strain WH8102. Different from WH7803, WH8102 swims in the liquid environment without using any external motility organelle like flagella. Also, the images are not provided here, based on the observation in fresh water motile cyanobacteria, the colony shape on the agarose surface is largely dependent on the agarose concentration and even wetness of the plate or even humidity in the air.

Line 212-213: no evidence was provided for the statement of the cell aggregation state, if possible, please provide microscopy images, small cell aggregates is difficult to see with naked eyes.

Line 301: no strong evidence is provided in this study to demonstrate that *Synechococcus* sp. WH7803 is non-motile. Maybe it is non-motile on the surface (just because we don't have the proper method to detect it), but it does seem to have some kind of motility in the liquid based on your results.

Minor:

Line 248 and 252: "medias" should be changed to "media"

Yiling Yang

Institute of Hydrobiology
Chinese Academy of Sciences
Wuhan, China

We thank the reviewers for their constructive comments. Following the reviewers' suggestions, and despite COVID restrictive protocols, we have been able to provide additional experiments to demonstrate: (i) the lack of cell-to-cell aggregation, (ii) the identical cell size and fluorescence between wild type and mutant strains, (iii) quantification of the sinking and positioning of cells in nutrient hotspots, and (iv) how pili structures, as well as encoded in their genomes, are actively produced in other picocyanobacterial strains. A point-by-point answer to reviewer's comments can be found below and changes in the text are highlighted in red in the 'marked up' version of the manuscript. We sincerely hope we have adequately addressed all of the points raised by the reviewers and the revised manuscript is now suitable for publication in Nature Communications.

REVIEWER COMMENTS

Reviewer #1 (Remarks to the Author):

This manuscript presents interesting data on the distribution of genes encoding components of the type IV pilus in marine cyanobacteria. This was a so far weakly studied research subject and requires definitely more attention. The authors show that many pilA-like genes are widely distributed among *Prochlorococcus* and *Synechococcus* species based on metagenomic and metatranscriptomic data and that also other components of a typical type IV pilus machinery are encoded by many marine cyanobacteria. Then they have used a genetically tractable marine organism, *Synechococcus* sp. WH7803, to perform functional studies and demonstrate that a generated mutant lacking pili shows a phenotype related to the performance of these planktonic strain in a water column. The main conclusion from these data is that type IV pili help marine cyanobacterial strains to stay suspended in a water column and to avoid sinking. Further, they have evidence that pili are important to evade predation by grazers.

I have reviewed a previous version of this article and surely the authors improved it substantially in accordance with the previous comments of the reviewers. However, the manuscript still suffers from insufficient knowledge of the current literature on T4P and understanding of the key biological concepts of the function of this nanomachine. Therefore, several conclusions made by the authors cannot be drawn based on the presented data which do not rule out alternative explanations. In addition, the study still lacks important control experiments and proper statistical analysis.

Authors: We thank the reviewer for the previous and current comments on our manuscript which have helped improve the presentation of our findings. We hope the revised manuscript and answers we provide below addresses these concerns.

Major comments:

1. line 55 and elsewhere (e.g. lines 92-95): In my opinion the authors have a conceptual misunderstanding of the pilus machinery and its function. It seems that the major pilin is PilA1 and all the other pili-like proteins are minor pilins. There is most probably nothing like a PilA1-like pilus or a PilA1-PilE, PilA2-PilV or PilA3-PilW pilus. As the authors detected only PilA1 and PilE in the exoproteome, it is likely that these pili subunits from the filament seen in EM. For the other so called minor pilins it is known that they can have various functions, such as initial assembly of the pilus, forming pilus tip, signaling function or DNA binding. Whether these other

pilins are really part of a distinct filament or part of the PilA1 filament or have other functions is complete speculation. Usually, the minor pilins do not form a whole filament, instead they fulfill functions related to the function of the major pilus. Therefore, without further experimental data nothing can be speculated about these genes.

Authors: The three pili 'types' we observe (and name) in marine *Synechococcus* sp. WH7803 are based on homology searches with those previously described in the well-known *S. elongatus* strain PCC 7942 (e.g. in Taton et al. 2020 Nat. Commun.). In Taton et al 2020 they observe that PilA1 is not necessary for the competence conferred by the PilA3-PilW pair suggesting PilA1 is likely not a part of the PilA3-PilW structure. Furthermore, Nagar et al. 2017 Env. Microbiol. show that neither PilA2 – for which a function has not been yet assigned – nor PilA3 are required for the planktonic phenotype provided by PilA1 pili. Additionally, it has already been shown that one same organism can produce different pili types from different PilA proteins, e.g. Neuhaus et al. 2020 Nat. Commun. (doi.org/10.1038/s41467-020-15650-w) show that *Thermus thermophilus* produces two different pili types using PilA4 and PilA5. In any case, we agree with the reviewer that the function of PilA2 and PilA3 has not been proven in this marine strain and that further work is needed. We have deleted any reference to 'PilA1-type pili' and the text now reads (line 97): “We propose that the genetic cluster encoding the six pilin-like proteins (Fig 1C and 1D) may provide three distinct pili functions. Based on homology with the annotated genes from *S. elongatus*¹⁶ and conserved domains found using the CD-search tool in NCBI, we suggest the three pilin pairs: PilA1-PilE, PilA2-PilV and PilA3-PilW (Fig 1C). Of these, shotgun proteomic analyses have only ever detected PilA1-PilE^{19,20} implying these are responsible for the pili observed in Fig 1A, although PilA2 was also detected in low abundance in cellular –but not extracellular– proteomic datasets²².”.

2. A major problem is the construction of the mutant. They found the downstream protein SynWH7803_1797 was also decreased in the proteome of the pilA1-pilE mutant. It is not clear, whether this is a polar effect of the mutation within one operon.

Generally, what is needed to be able to make clear conclusions about the mutant study is:

- i) To show transcriptomics data to give a hint which genes are co-transcribed in this region of the genome and where is a possible promoter.
 - ii) Complementation of the mutant with pilA1 and pilE to discriminate between the role of both pilin genes and to exclude polar effects on downstream or maybe also upstream genes.
- In general, I find it dangerous in bacterial genetics to work with only one clone without successful complementation of the phenotype.

Authors: The slight decreased abundance of the hypothetical protein SynWH7803_1797 (3.2x) in the pili knock-out mutant is possibly due to a polar effect as the analysis of one of our transcriptomic datasets suggests it is part of a *pilA1-pilE* operon (data not shown). Interestingly, though, the production of SynWH7803_1797 was obviously not abolished and the protein was still detected in the cellular and exo-proteomes of the pili mutant strain carrying out whichever its function. Thus, it is unlikely involved in the unambiguous pili abolishment and drastic sinking phenotype in the mutant strain.

We agree that complementation would be fantastic if feasible. Unfortunately, to date, there is no known plasmid that will reliably replicable in marine *Synechococcus*, preventing complementation. While genetic manipulation in *Prochlorococcus* remains impossible (Laurenceau et al. 2020 Access Microbiol.), we can generate knockout mutants in marine *Synechococcus* sp. WH7803 with an improved protocol provided as supplementary information

in this manuscript. Nevertheless, even when everything goes to plan, it takes over 3 months to generate a mutant strain (formation of colonies in sloppy agar take 4-6 weeks) highlighting that, while possible, it is still challenging.

In any case, and as mentioned in lines 364 of the methods section, while generating the *pilA1-pilE* mutant we picked three independent transconjugant colonies and clearly observed a loss of the planktonic form in all independent mutants. For practical reasons we only made axenic and carried on working with one of these mutants.

Considering all information available: i) the high similarity between the wild type and mutant proteomes (suggesting negligible polar effects), and ii) the similar phenotype described in non-piliated *S. elongatus* (Nagar et al. *Env. Microbiol.* (2017)) and *Prochlorococcus* sp. MIT9313 (this study), makes us confident that the pili are, in fact, responsible for the observed phenotype.

3. Fig. 1A and Fig. S1: It seems that the mutant is smaller than the WT. Is this statistically relevant and do you see other obvious phenotypes in e.g. pigmentation or growth? It would be really beneficial to have a more comprehensive characterization of the mutant in order to exclude that the detected effects (related to sinking and grazing) are really due to the lack of pili and not because of size difference or different accumulation of carbohydrates. It is known from other cyanobacteria that the inactivation of PilA1 might have a pleiotrophic phenotype. Interestingly, there are only very small changes in the proteome of the pili mutant which does not hint to a major effect of the *pilA1-pilE* deletion on other cellular functions. However, changes can occur also on the level of protein activity controlled e.g. by second messenger molecules.

Authors: As the reviewer points out, the proteomic differences between wild type and pili deficient mutant are remarkably small suggesting, presumably, very small metabolic variations between both strains. The difference in cell size and/or aggregation was further explored by fluorescent microscopy and flow cytometry (Supplementary Fig. S3 and S4) revealing remarkable similarities between wild type and mutant strains. Interestingly, even non-shaken cultures of the pili mutant, in which cells sedimented to the bottom of the culture flask, presented no obvious sign of cell aggregates or major cell size variations. The additional information has been added to the main text (line 224): “*We further confirmed no apparent differences in cell size between the mutant and wild type strains (Fig S3 and S4), and observed no cell aggregates when grown in shaken or non-shaken liquid cultures (Fig S4).*”

I have a further problem with the EM figures. As I have pointed out already in my last review on this paper, this marine *Synechococcus* strain is known to be around 1 – 1.5 micrometer in size, here it seems that it is at least 4 micrometer which would be the size of *Synechococcus elongatus*. Whereas, the mutant has roughly the correct size of what you would expect for this organism. Do you have any explanation?

Authors: There is no doubt the EM images provided are of *Synechococcus* sp. WH7803 because all molecular analyses (e.g. PCR, sequencing and proteomic analyses) performed on these same cultures always match this strain. Furthermore, the pili observed by EM are also detected in very high abundance by exoproteomics in every analysis performed with *Synechococcus* sp. WH7803. The reason for the difference in size may be that cells are prior to division or a slight difference in culture stages. As observed in the additional microscopy images and flow cytometry analyses (Supplementary Fig. S3 and S4), there seems to be no substantial difference between wild type and pili mutant cell sizes.

4. Line 251: Are there any transcriptomic data which would show that *Prochlorococcus* can really regulate the production of pili in response to different media? Maybe it is sedimentation (the surface contact) which controls pilus production.

Authors: There are four transcriptomic studies available for *Prochlorococcus* sp. MIT9313:

- Thompson et al. ISME J 5, 1580–1594 (2011), where RNA micro-arrays were performed to analyse the transcriptional response of this strain to iron depletion. Genes of the pili apparatus are not reported in the manuscript or supplementary data.
- Voigt et al. ISME J 8, 2056–2068 (2014), where the transcriptome of this strain was analysed for antisense and non-coding RNAs. As well as a lack of conditions to compare, the supplementary data does not allow to assess the abundance of transcription for *pilA1* or other related genes.
- Aharonovich and Sher, ISME J 10, 2892–2906 (2016), where transcriptomes of this strain were compared between axenic and heterotroph co-culture conditions. *pilA1* and *pilT* were not differentially expressed between axenic and co-culture conditions.
- Fang et al. Env Microbiol 21, 2015–2028 (2019), where the transcriptomes of this strain were compared between control (Pro99 media) and media where viral lysate had been added. While *pilA1* was abundantly expressed (as we observe in our study), there was no significant differences between the conditions tested.

The lack of adequate datasets makes it hard to evaluate the regulation of pili synthesis in this strain although this is an excellent suggestion for future work. Our proteomic data clearly correlates the presence of PilA1 in the exoproteome with the planktonic form of the strain.

Surface contact is unlikely to be causing the inhibition of pili production because the supernatants were obtained from shaken cultures where cells were maintained in suspension. Future work will allow us to evaluate how each strain regulates pili production, and this may well be strain-specific. We believe this is out of the scope of the work we present here but will definitely take it into account in future studies.

Line 268: again, “... where it remained via the production of pili.” This is an over interpretation of the data. You did not show that this is due to the production of pili. This could be also a secondary effect of lack of pili. It is e.g. shown that mechanosensing via type IV pili induces a signal transduction chain via second messengers which leads to activation of different cellular responses. Further, large amounts of PilA1 can be also detected in the exoproteome when pilus retraction is disturbed (e.g. *pilT* mutant strain). That means that you still can detect PilA, even if the pilus might be functionally inactive. In general, the authors should be more careful about what they have really measured and what can be concluded from these measurements.

Authors: We agree this is speculation and have now indicated it in the text (line 283): “...where it remained, presumably, via the production of pili...”

5. What is the diameter of the pili? Does it fit to a typical type IV pilus? This should be measurable from the EM pictures.

Authors: The diameter was measured via ImageJ giving 6 nm. This information has now been added to the text (line 64): “...this marine picocyanobacterium presented multiple pili of similar thickness (diameter of ~6 nm), each ~10 μ m in length.” And in the Methods section (line 375): “Pili length and diameter were measured using ImageJ (version 1.53a).”

6. Figure 1E: Several aspects in this figure are not correct: PilB and PilT are hexamers. PilA are localized in the membrane before they are incorporated into the pilus structure. They are not cleaved in the cytoplasm. The picture is here completely misleading. PilC is a single subunit complex. Didn't they detect PilM? What is with PilT1, they show PilT2 and PilT3. On which base a PilT3 protein was assigned?

Authors: We are grateful for pointing out these issues. They have now been amended in the figure and text. PilM was not detected and two copies of *pilF*, *pilO* and *pilP* are present in the *Synechococcus* sp. WH7803 genome but only the one with the highest homology to *S. elongatus* sp. PCC7942 is represented. While *S. elongatus* contains 3 different PilT copies, *Synechococcus* sp. WH7803 only has 2 copies. All this information is now indicated in the legend to Fig 1: “PilM was not detected (n.d.). Searches of *pilF*, *pilO* and *pilP* (highlighted with an asterisk) retrieved two homologues in *Synechococcus* sp. WH7803 and only those with highest homology to *S. elongatus* PCC 7942 are shown. While *S. elongatus* PCC 7942 encodes three *pilT*, only two were found in *Synechococcus* sp. WH7803.”

7. In my opinion, the authors should try to quantify the data shown in Figure 4.

Authors: The data from Fig 4B (microscopy tracking) was previously quantified and data is available in Supplementary Table S2, and sinking rates were measured as indicated in the text (lines 219-221).

The experiment shown in Fig 4D has now been repeated and fractions quantified by flow cytometry. Data has been included in Supplementary Fig S6 and the associated figure legend reads: “Fig S6. Retention of pili-producing *Synechococcus* in nutrient-rich fractions within the water column. Nutrient layers were set up as previously described in Fig 4D and experimental setup is depicted on the left hand panel. Syringes (10 ml) were used to facilitate the harvesting of the seven fractions. Cells from *Synechococcus* sp. WH7803 (WT, red) and pili-mutant cultures (Δ pili, blue) were diluted to $\sim 10^7$ cells/ml in oligotrophic seawater and placed on top of the column. 100 μ l were harvested from each one of the fractions (~ 1 ml) at an initial time point and at days 2 and 4. Cells were quantified by flow cytometry. Error bars represent the standard deviation from three independent columns.” We have also referenced this new data in the main text (line 284) and in the Fig 4 caption: “Fig S6 shows the quantification of the sinking and accumulation of cells in the different layers over time.”

Minor comments:

Line 42: *Synechococcus elongatus* also performs phototaxis using type IV pili and most probably PilA1 (Yang et al., 2018).

Authors: While it is true that *S. elongatus* UTEX 3055 shows phototaxis (Yang et al 2018), in the cited study they do not prove the twitching motility was performed via pili and much less by PilA1. This is just speculated in the Discussion of this paper: “Notably, neither operon is required for cell motility, which is most likely mediated by peritrichous type-IV pili based on the moving

behavior of UTEX 3055 on the agarose surface (Movie S7) and the observed distribution of pili in PCC 7942 (52).” Hence, we cannot be confident of such a statement to be included in this section.

Line 44: I think it is not quite correct to say that PilA3 performs acquisition of DNA, it was only shown that it is needed for transformation, but the exact function is unknown to my knowledge.

Authors: This has been reworded as suggested by the reviewer (line 45): “*While the third of three PilA-like proteins (PilA3) encoded by *S. elongatus* is required for transformation¹⁶...*”

Line 106 and elsewhere: the authors mix up the terms pilin and pilus. Here, you most probably mean the pilin. You can’t say from genomic data whether a pilus is present. You can only say whether the genes encoding pilus subunits are present in genomes.

Authors: We have changed this accordingly (line 112): “*In *Synechococcus*, pilA1 was prevalent in all clades...*”

Line 117 (see also major comment 1): You did not show that there is something like a PilA3-type pilus. You also did not show that this minor pilin can bind DNA. To my knowledge, it is also not possible to predict functions of these minor pilins from sequence. However, the authors did this in a large extent. Anyway, it seems the similarity between e.g. PilA3 from *Synechococcus elongatus* and the 1790 protein sequence from WH7803 is so low that I wonder on which base you think they have similar functions. The same is also true for the assignments of the other pilin.

Authors: We agree and highlight in the text that further work is needed to confirm this (lines 122). We would not have suggested this hypothesis if we had not found such a nice correlation between the presence of the *comE* genes encoded by all strains with a *pilA3*-like pilin. PilA3-PilW and ComE have recently been described to be involved in DNA uptake in *S. elongatus* (Taton et al. 2020 Nat. Commun.) and, hence, this may well occur in marine picocyanobacteria containing these competence genes (*i.e.* *comE* genes).

255: better PilA1 accumulation, you do not measure pili production

Authors: The text has been modified (line 269): “*We further explored the effect of nutrient stress on the detection of PilA1 in *Synechococcus* sp. WH7803 exoproteomes...*”

260: you do not detect pili, this you can do e.g. by EM

Authors: The text has been modified (line 276): “*PilA1 detection became most obvious under higher nutrient conditions (60-fold increase in PilA1 at 88 μ M N and 1.8 μ M P when compared to nutrient deplete seawater; Table S6).*”

296: There is nothing like a PilA1-type pilus (see comments above)

Authors: The text has been modified (line 315): “*This ~~PilA1-type~~ pilus, widely distributed and expressed amongst dominant marine oligotrophic cyanobacteria...*”

299: I think there is no evidence, that this regulation is tight and it is unclear what kind of regulation is meant here. There are no data on that.

Authors: We agree. We have now modified the text (line 317): *“This mechanism responds to discrete stimuli e.g. nutrients...”*

302: Can you exclude that picocyanobacteria which harbor the genes for T4P are nonmotile?

Authors: This is a very interesting point. Initially, we observed that the model strain used to study motility in these cyanobacteria, *i.e.* WH8102, did not encode PilA1 (or any of the other pilin proteins) despite encoding the transmembrane mechanism. Nevertheless, the other sequenced motile strain WH8103 – which also encodes the swimming proteins SwmA and SwmB – *does* encode *pilA1* (see Fig 2) and, hence, the motility mechanism does not exclude having the ability to produce pili. We have now modified the text to highlight pili are not exclusive to non-motile picocyanobacteria (line 319): *“Therefore, as opposed to flagellated bacteria that show positive chemotaxis towards nutrient hotspots in the oceans³², non-motile picocyanobacteria may apply a more passive strategy which consists of elongating their pili when they encounter preferable conditions to remain in an optimal position while drifting in a water body.”*

313: Can you provide citation which prove that the genomes marine strains do not encode components of chemotaxis systems?

Authors: To the best of our knowledge, there are no conventional chemotaxis systems in the marine picocyanobacterial genomes sequenced to date. Nevertheless, there is an old paper (*i.e.* Willey J.M Waterbury J.B (1989) *Chemotaxis toward nitrogenous compounds by swimming strains of marine Synechococcus spp.* Appl. Environ. Microbiol.55, 1888–1894) which describes how one motile *Synechococcus* strain WH8113 presented chemotaxis towards nitrogenous substrates. Unfortunately, the genome of this strain has not been sequenced and the mechanism has not been identified. In any case, there are no references available highlighting that marine cyanobacterial genomes lack conventional chemotaxis systems.

Fig. S3B: what is platocyanin? Do you mean plastocyanin?

Authors: It is plastocyanin. This has been amended.

Fig. S2: Please, check the legend, I see only models for PilA1, not for PilA2, PilA3 and PilA.

Authors: This has been amended.

Reviewer #2 (Remarks to the Author):

The authors present genetic evidence of the encoding of a type IV pilus by the abundant marine picocyanobacteria *Synechococcus* and *Prochlorococcus*. They applied culture experiments using *Synechococcus* isolates of wild-type (WT) and mutant strains lacking the pili that suggested that these proteins may increase a cell's viscous drag and aid in avoiding predation in the natural

environment.

This is a comprehensive and very nicely presented study, and I found the genetic work particularly enticing and elegant, complemented by genomic analyses of known marine cyanobacterial isolates and single cell genomes. The expression and distribution of pilA1 in the TARA metagenome and metatranscriptomes provided a better understanding of the distribution of these pili proteins in the natural environment, suggesting an ecological role. The methodology to produce *Synechococcus* mutants lacking pili proteins developed in this study should prove helpful for further studies of the physiological and biophysical mechanisms underlying the production and role of these pili in the natural environment.

I mainly have issue with the ecological interpretations and conclusions from the culture work, which in my view are too far reaching and not convincingly supported by biophysics nor the results of the experiments comparing WT vs mutant strains. I detail these below:

Authors: We thank the reviewer for the kind comments and constructive review provided below.

1. The difference in daily sinking rate of the pili mutant compared to the slight uplift is very small (Line 207 and following). While this may make difference in a test tube, in the ocean, both would be considered non-sinking. Also, in Fig. 4, the sinking rate of 8.4 mm/day determined from the tracking experiments would not explain the accumulation of the majority of pili mutant cells on the bottom of the tube after three days. Alternatively, the pili mutants might be physiologically different and show an increase in sinking rate. For example, the lack of pili could allow for more cell-to-cell contact and aggregate formation, thus enhancing sinking rates.

Authors: Due to their size and presumably slightly higher density than seawater, it is perhaps not surprising for single *Synechococcus* cells to sink at ~1 cm/day. Whilst in the context of a single day this may sound insignificant, over longer (e.g. annual) time scales this would mean cells would remain in the euphotic zone much longer and not having to rely on erratic uplifting currents, which would be especially difficult in strongly stratified systems. At the micro-scale, this would mean cells are able to remain longer in 'optimal spots' (e.g. of light, nutrients, etc) and hence achieve higher division rates without the need for active motility motors such as flagella. Note also that production of this pili structure also has the added bonus (and hence biological significance) of allowing cells to avoid being grazed.

We do agree with the reviewer though that there is a 'slight uplift' which we believe is not due to an experimental artefact (*i.e.* there was an uplift in almost all of the replicates analysed) and this definitely deserves further investigation. With this in mind we now emphasise that "*pili did not appear to confer apparent twitching motility*" (line 222) to not confuse with swimming motility.

In terms of cell aggregation, we have carried out more experiments to visualise and quantify this phenomenon in our cultures. We have visualised by flow cytometry (additional Supplementary Fig S3) and fluorescent microscopy (additional Supplementary Fig S4) both wild type and pili mutant cultures with no apparent formation of cell aggregates. We even examined non-shaken cultures of the pili mutant, *i.e.* where cells sink to the bottom of the flask and thus come into very close cell-to-cell contact and, still, did not observe aggregates (Supplementary Fig S4). Interestingly, though, at higher metal concentrations (e.g. 5x the metal concentration normally added in ASW medium), cells tended to clump and rapidly sink within minutes and exoproteomic analyses show a 50x PilA1 reduction in high metal exposed cultures. This may be due to a flocculation or toxicity effect that requires further investigation (data not shown).

2. Similarly, the accumulation of WT cells in the nutrient rich layer of the tube in Fig. 4D could simply be due to an accumulation of cells that exhibit an increased growth rate. And it seems like some cells still sink beyond the ESW layer. Ultimately, the authors need to develop a biophysical explanation, e.g., using the Navier-Stokes equation, on the mechanism how pili could increase drag in a regime where $Re \ll 1$.

Authors: Fig 4D has now been repeated and quantified in Supplementary Fig S6. We can confirm that both wild type and pili mutant cells doubled by the end of the experiment on day 4 (Supplementary Fig S6). As observed in Fig 4A and 4D, and still today while we routinely sub these cultures, the phenotype is extremely clear. The pili mutant has never returned to its planktonic phenotype.

According to the standard laws of low Reynolds number hydrodynamics, pili have to increase the overall drag of a piliated cell over that of a naked cell. This is a consequence of two facts: 1) the existence of pili alters the flow field that would otherwise be generated by the sedimentation of a naked cell; 2) the Helmholtz minimum dissipation theorem, stating that the steady Stokes flow of an incompressible fluid has the smallest rate of dissipation than any other incompressible motion with the same boundary velocity. Taken together, these two points imply that adding pili to a naked cell will increase the dissipation of the flow resulting from its translation. Correspondingly, this implies that the drag coefficient will increase. Unfortunately, a *precise* estimate of such an increase is complicated, and would rest on a series of assumptions regarding the mechanical properties of the pili for which we do not have a precise physical characterisation. Nevertheless, thanks again to the minimum dissipation theorem, it is still possible to obtain an upper bound for the drag increase. At low Reynolds number, the drag of an object cannot be larger than that of the smallest circumscribed sphere. If the cell is roughly spherical with radius “ r ” and the pili are uniformly distributed around the cell with length “ l ”, and they are stiff enough that they don’t bend due to the sedimentation (a measurement of pili stiffness is lacking), the drag coefficient cannot be reduced more than a factor $r/(r+l)$. Taking $r=1\mu\text{m}$ and $l=10\mu\text{m}$, one gets a factor of approximately 10, *i.e.* a sedimentation of $\sim 1\text{mm/day}$.

3. Also, when checking pilus distribution among picocyanobacteria, in clades II and II the gene was less abundant. However, these clades are the most abundant in the Sargasso Sea for example (Ahlgren and Rocap 2012), so one wonders why those clades do not have this gene if it is deemed so important for maintaining suspension in the water column. This appears inconsistent with the arguments presented regarding the importance of the pili proteins in these abundant pico-cyanobacteria.

Authors: We agree that, like clades II and III, many *Prochlorococcus* strains also lack pili and this is reflected, to some extent, in our interrogation of the TARA datasets (Fig 3). In Fig 3 we show, excluding Southern and Arctic Oceans, that the Atlantic Ocean presents the lowest abundance of picocyanobacterial *pilA* amongst other oceans which could be attributed to the different nutrient limitations between oceans. We suspect that, like the swimming proteins SwmA and SwmB which are exclusively found in *Synechococcus* strains from clade III, and the pilus we present in this study, marine cyanobacteria may have also evolved other mechanisms for remaining planktonic that require further investigation. The ecological significance of their distribution is definitely something that deserves further investigation and that goes beyond the current study.

4. One of the abundant strains of *Synechococcus* (WH8102) has been found to produce transparent exopolymeric particles (TEP) in culture, increasing stickiness of the cells and leading to formation of cell aggregates (Cruz et al., 2019). This could interfere with the postulated effect of the presence of the pili in decreasing sinking rate, and should be discussed.

Authors: Further discussion has been added based on the additional data included in Supplementary Figs S3 and S4. The text now reads (line 282): *"While the pili mutant, as expected, sank through the gradient independently of nutrient availability, the wild type strain was able to position itself in the nutrient-replete layer where it remained, presumably, via the production of pili (Fig 4D and Fig S6). Sinking was confirmed to not be caused by cell aggregation as previously reported (Cruz et al., 2019) nor to a variation in cell shape and size (Fig S3 and S4)."*

5. Regarding the grazing experiments: previous studies on the role of surface proteins have already shown inhibition of grazing on *Synechococcus*. Strom et al., 2012 found that mutant cells of Syn WH8102 lacking the SwmB protein were more susceptible to being grazed compared to the WT, which could be parallel to what was found herein with pili-lacking mutants, and needs to be discussed.

Authors: We have added information on this reference (line 295): *"Given the cell surface nature of this pilus and previous reports showing grazing inhibition by a giant cell surface protein (i.e. SwmB in Synechococcus sp. WH8102) (Strom et al 2012), we assessed whether these cell appendages could mediate ecological interactions with other organisms. Thus, Synechococcus sp. WH7803 and the pili mutant grown..."*

Minor comments:

1. The methods are clearly written and referenced by supplemental material, detailed enough to allow for reproducibility, especially for production of *Synechococcus* mutants. However, to make the nutrient step gradient experiments more reproducible (lines 359-362), the authors could present more details on the addition of the sucrose. Was it added just to create the different density layers? This needs to be clarified.

Authors: Yes, the sucrose was added to create the nutrient step gradient. This is now clarified (lines 379): *"The movement of wild type Synechococcus sp. WH7803 and pilus mutant through a nutrient step gradient was performed by placing washed cells in oligotrophic SW on top of a water column. Nutrient deplete (oligotrophic SW) and nutrient amended layers (ASW) were placed below the top cell-containing layer as indicated. Density separation between layers was achieved by an increasing sucrose concentration (i.e. 2.5% w/v per layer)."*

2. Line 46: the wild-type of *S. elongatus* was already a biofilm former; the referenced study showed that removing the pili didn't stop biofilm formation, just occurred more slowly - the cells were not planktonic even in wild type.

Authors: We disagree. To our knowledge the wild type *Synechococcus elongatus* PCC 7942 has its biofilm development machinery 'inhibited' and, only when PilA1 pilus is inhibited, the strain forms biofilms (not so much when PilA2 is inhibited). This is the text extracted from Nagar et al 2017: *"We previously reported that the inactivation of Synpcc7942_2071 of the cyanobacterium Synechococcus elongatus PCC 7942 results in biofilm development in an otherwise planktonic"*

strain (Schatz et al., 2013). The formation of a bona fide biofilm by the mutant strain indicates that a genetic program that underlies biofilm development exists in the laboratory strain and is constitutively inhibited." This is also backed up by the publication from the same group Schatz et al 2013 (doi: 10.1111/1462-2920.12070): "We identified a mutant of the cyanobacterium *Synechococcus elongatus*, which unlike the wild type, developed biofilms. This biofilm-forming phenotype is caused by inactivation of homologues of type II secretion /type IV pilus assembly systems and is associated with impairment of protein secretion." The protein they are referring to is PilB which elongates the pili (mainly by assembling PilA1).

3. Line 68: modify to "N-terminal"

Authors: This has now been modified (line 70).

4. Line 332: what is the meaning of "CCAP 1900/1 cells"?

Authors: This refers to the strain. We have now reworded this for clarification (line 351): "...and *Cafeteria roenbergensis* (strain CCAP 1900/1) cells."

Reviewer #3 (Remarks to the Author):

Overall comments

The manuscript entitled "Pili allow dominant marine cyanobacteria to avoid sinking and evade predation" by Aguilo-Ftrretjans et al deal with the function of type 4 pili. They found that type 4 pili encoding genes are present in about one quarter of picocyanobacteria. The authors claim that pilus filaments confer two biological important functions to marine picocyanobacteria, one is to allow cells to float at nutrient rich region in the water column, and another is to avoid predation by grazers. These findings are novel, and I believe they are of great interest to cyanobacterial field and maybe to a broader field of ecology.

The manuscript is clearly written, and the important experimental details are provided. However, some statements in the text are not well supported.

Authors: We hope all concerns are addressed by the additional experiments requested and which have now been included in this revised version of our manuscript.

My major concern is the experimental evidences that support these conclusions are based only on the model organism *Synechococcus* sp. WH7803. To make a general conclusion for marine picocyanobacteria, the author should perform the same sinking experiment and grazing experiment with 2-3 more piliated strains.

Authors: Apart from the data generated for *Synechococcus* sp. WH7803, we also provide a clear correlation between the planktonic phenotype and the presence of PilA1 in the *Prochlorococcus* strain MIT9313 (Fig. 4C and Suppl. Table S4). Moreover, a similar phenotype was shown for the non-piliated mutant of *Synechococcus elongatus* PCC 7942 (Nagar, E. et al. 2017 Environ. Microbiol.) strengthening our conclusions. Additional proof for other strains would require to genetically knockout these genes, remaining impossible in *Prochlorococcus* (Laurenceau et al 2020 Access Microbiol.). While feasible in marine *Synechococcus*, the generation of mutants is a

long laborious and challenging task, taking over three months when everything goes to plan because these marine cyanobacteria take 5-6 weeks to grow colonies in sloppy agar. We have, though, generated additional proof that, as well as encoded, pili are actively produced in other strains (i.e. *Synechococcus* sp. WH7805; which has now been included as panel B in the supplementary Fig S1) and report PilA1 in the exoproteome of *Synechococcus* sp. BL107 (Christie-Oleza et al 2015 Env Microbiol.). We have implemented this new evidence in the main text (line 125): “As well as encoded in their genomes, we further confirmed PilA1 was actively produced in other picocyanobacterial strains i.e. *Synechococcus* sp. BL107 (8.9% of its exoproteome¹⁹) and *Prochlorococcus* sp. MIT9313 (see below). Furthermore, pili similar to those observed in *Synechococcus* sp. WH7803 were actively assembled in *Synechococcus* sp. WH7805 (Fig S1) despite the low homology between their pilA1 genes and absence of a pilE homologue in the later strain (Fig 2).” The images of the pili in *Synechococcus* sp. WH7805 have been added in Fig S1.

Major comments:

Figure 1:

1. Why all the pili mutants (also from Figure S1) seem to have a smaller cell size?

Authors: We have now visualised wild type and pili mutant cultures via flow cytometry (Supplementary Fig S3) and fluorescence microscopy (Supplementary Fig S4), and observed no apparent differences in size, morphology or auto-fluorescence. This new data has been provided in the new version of our manuscript (line 224): “We further confirmed no apparent differences in cell size between the mutant and wild type strains (Fig S3 and S4), and observed no cell aggregates when grown in shaken or non-shaken liquid cultures (Fig S4).”

2. The drawing for pili could be misleading. PilD is a membrane protein, not cytoplasmic; PilB and PilT are hexamer, not octamer; PilM is missing and there is no mention of this protein at all throughout the manuscript, is it because there is no PilM homolog in WH7803? How conserved is this protein in bacterial type 4 pili?

Authors: All these issues have been addressed in the new version of Fig 1E. No homologue of PilM was found in the genome of WH7803 and is now indicated as ‘n.d.’ (non-detected) in the Table of homologues with *S. elongatus* of Fig 1E and in the figure legend: “PilM was not detected (n.d.). Searches of pilF, pilO and pilP (highlighted with an asterisk) retrieved two homologues in *Synechococcus* sp. WH7803 and only those with highest homology to *S. elongatus* PCC 7942 are shown.”

Figure 4:

1. Could the authors perform the cell sinking experiment for a different piliated picocyanobacterial strain if they can find one? So that you can consider it a general mechanism but not just exist in WH7803.

Authors: As mentioned above, we showed the same phenotype in *Prochlorococcus* sp. MIT9313 (Fig. 4C). Equally, the phenotype was also reported for *Synechococcus elongatus* PCC 7942 (Nagar et al. 2017 Environ. Microbiol.) and, hence, we are confident of the phenotype given by the pili in planktonic cyanobacteria.

2. Based on the movies 1-2, mutant cells seem a little bit longer than the WT (but the TEM images show a smaller cell size of the mutant). Since the cell size/shape may affect the sinking rate, I think it is necessary to compare the mutant cell size to the wild type, to make sure that pili mutant does not affect cell size.

Authors: This has now been confirmed via flow cytometry (Supplementary Fig S3) and fluorescent microscopy (Supplementary Fig S4). Despite the similarity of cell size and shape, wild type cultures consistently retain a planktonic form and pili mutant cultures always sediment to the bottom of the flask, whichever the culture's growth phase. We further confirm in this revised version of the manuscript that this is also not due to cell aggregation (fluorescent microscopy images in Supplementary Fig S4).

3. I did not find in the material and method that how the cells are labelled for the tracking experiment in panel B and movies 1-4.

Authors: Cells were not labelled and were tracked via dark-field microscopy. The methods have been detailed in lines 396.

Figure 5:

1. Likewise, it would be more convincing if the authors can show that the same result is obtained with another piliated picocyanobacterial strain.

Authors: We believe the phenotype observed in these experiments is clear. In order to perform these same experiments with other strains we would need to generate pili-deficient mutants which in our hands has not proved possible in other piliated strains e.g. WH7805.

Line 63: how do you get the number of 10µm length for the pili, it is hard to see the end of a pili from the TEM images, they seem all twisted around each other, which make it difficult to trace.

Authors: By measuring pili using ImageJ we observed pili over 10 µm in length. This has now been detailed in the methods section (line 375): "*Pili length and diameter were measured using ImageJ (version 1.53a).*"

Line 90-91: you stated that you find all components necessary for pilus assembly, but I don't see the structural protein pilM, which I think is important for the pili function.

Authors: True. We are unsure if it is completely lacking or whether there is an alternative protein that carries out the function of PilM. We have now highlighted this in the legend of Fig 1: "*PilM was not detected (n.d.). Searches of pilF, pilO and pilP (highlighted with an asterisk) retrieved two homologues in Synechococcus sp. WH7803 and only those with highest homology to S. elongatus PCC 7942 are shown.*"

We have also modified the main text (line 97): "*we were able to find all components necessary for pilus assembly in Synechococcus sp. WH7803 except for pilM (Fig 1E).*"

Line 210-212: the conclusion of "pili did not appear to confer motility" is not appropriate here purely based on the comparison of the colony shape with the Synechococcus sp. strain WH8102. Different from WH7803, WH8102 swims in the liquid environment without using any external motility organelle like flagella. Also, the images are not provided here, based on the observation

in fresh water motile cyanobacteria, the colony shape on the agarose surface is largely dependent on the agarose concentration and even wetness of the plate or even humidity in the air.

Authors: We agree and have clarified we were referring to twitching motility and not swimming motility. The text was modified (line 222): *“pili did not appear to confer apparent twitching motility, with both mutant and wild type strains producing ‘pin-prick’ colonies in sloppy agar plates as opposed to fuzzy and expanding colonies characteristic of motile⁸ and twitching strains¹⁴.”*

Line 212-213: no evidence was provided for the statement of the cell aggregation state, if possible, please provide microscopy images, small cell aggregates is difficult to see with naked eyes.

Authors: Flow cytometry data (Supplementary Fig S3) and fluorescent microscopy images (Supplementary Fig S4) has now been provided and the text now reads (line 224): *“We further confirmed no apparent differences in cell size between the mutant and wild type strains (Fig S3 and S4), and observed no cell aggregates when grown in shaken or non-shaken liquid cultures (Fig S4).”*

Line 301: no strong evidence is provided in this study to demonstrate that *Synechococcus* sp. WH7803 is non-motile. Maybe it is non-motile on the surface (just because we don't have the proper method to detect it), but it does seem to have some kind of motility in the liquid based on your results.

Authors: We agree that the slight uplift observed in the wild type strain may come as a result of some kind of motility that deserves further investigation. We have hence deleted 'non-motile' from this sentence (line 320).

Minor:

Line 248 and 252: “medias” should be changed to “media”

Authors: This has now been addressed as suggested by the reviewer.

Reviewers' Comments:

Reviewer #1:

Remarks to the Author:

The revised manuscript is significantly improved by some aspects, but retains significant deficiencies in others.

Importantly, they have weakened several of their previous strong and far-reaching statements and conclusion based on the limited data on the analysis of one mutant strain. I agree that genetic engineering of such a marine strain is a challenge and is by far not comparable to genetic methods applicable to freshwater cyanobacterial strain or even heterotrophic strains. The more one has to be very careful with strong conclusions. Therefore, in my opinion, the abstract and the concluding remarks are too definitive and imply stronger evidence for the suggested mechanism than provided. Alternative explanations are largely not discussed.

Further, my concern about the size of the cells is still somehow valid. The problem is that the reader will see the electron micrographs in Fig. 1A and S1 where it is very clear that the cells differ in size and then they will be guided to the supplement where flow cytometry and fluorescence microscopy imply that there is no difference. I do not know why the electron micrographs give such a misleading picture, but this whole story is quite unsatisfactory. Especially, because the size and shape of cells would have such a strong influence on all the following experiments.

My concern regarding the bioinformatics analysis of a potential pilus machinery and figure 1E has only partly addressed. First, I have to apologize that I stated in my previous review that PilC is a monomeric subunit. This is not true, though several schemes show PilC as a single subunit. Based on structural data in heterotrophs this is a dimer and the authors should have noticed this. In the current figure 1E now this pilus part looks really strange, because PilC monomer is shifted to the left, leaving some open space on the right. Further PilN and PilO should be visualized as transmembrane proteins in the picture. And I have serious doubts that PilM is missing. If you check the *Synechococcus* genome, there is a "standard" pilMNOQ (WH7803_2367-2364) operon. I agree pilM is not very well conserved, but if you do a hmmsearch with a model based on cyanobacterial PilM homologs you will detect a weak homology to the gene upstream of pilN. Homologs of WH7803_2367 are typically located directly upstream of pilNOQ genes as well. Obviously weak homology and synteny hint at the possibility that marine cyanobacteria evolved an atypical pilM. This should be discussed instead of writing that no pilM was detected. I am also not so happy with the naming of the 3 PilT homologs. Again, there is a pilCTB operon in the genome and an additional copy of PilT containing a P-rich N-terminus. The pilT gene in the operon should be named pilT1, and the other copy pilT2 in accordance with the designation in other model organisms. The pilT (2) and pilT (3) naming scheme in the current paper is quite confusing. Further, the statement that the "six pilin like proteins may provide three distinct pili functions" is much too speculative in my opinion.

Minor:

In the legend to Fig. 5 you still say that it is a pilA1 mutant, which is not correct.

Reviewer #2:

None

Reviewer #3:

Remarks to the Author:

The authors have performed adequate experiments to solve most of my concerns. The images in Fig.S1B has not-so-clean background and I am not sure if those shown are pili or not, especially when you don't see a complete cell in it. Other than that I have no more questions.

We again thank the reviewers for their constructive comments. Following the reviewers' suggestions, we have now measured the cell size of wild type and pili mutant cells as well as modified Fig 1 and supplementary Fig S1 as recommended by the reviewers. A point-by-point answer to reviewer's comments can be found below as well as the modified figures, and changes in the text are highlighted in red in the 'marked up' version of the manuscript. We sincerely hope we have adequately addressed all of the points raised by the reviewers and the revised manuscript is now suitable for publication in Nature Communications.

REVIEWER COMMENTS

Reviewer #1 (Remarks to the Author):

The revised manuscript is significantly improved by some aspects, but retains significant deficiencies in others. Importantly, they have weakened several of their previous strong and far-reaching statements and conclusion based on the limited data on the analysis of one mutant strain. I agree that genetic engineering of such a marine strain is a challenge and is by far not comparable to genetic methods applicable to freshwater cyanobacterial strain or even heterotrophic strains. The more one has to be very careful with strong conclusions. Therefore, in my opinion, the abstract and the concluding remarks are too definitive and imply stronger evidence for the suggested mechanism than provided. Alternative explanations are largely not discussed.

Authors: We are glad the revised version of our manuscript has significantly improved. As well as our data from the pili knockout mutant in *Synechococcus* sp. WH7803 (which still today, after years of culture subbing, has never recuperated its planktonic form), we have evidence that PiliA is *only* detected in planktonic cultures of *Prochlorococcus* sp. MIT9313, and that *Synechococcus elongatus* loses its floatability when pili production is abolished (Nagar et al 2017 *Env. Micro.*). This, together with our tracking data of wild type and mutant cells, and the ability of wild type cells possessing pili to evade grazers we believe is entirely consistent with the conclusions we present. Whilst alternative explanations are not clear to us we note the referees concerns with our definitive conclusions and have modified the final lines in the abstract as follows: "*The evolution of this sophisticated floatation mechanism in these purely planktonic streamlined microorganisms has important implications for our current understanding of microbial distribution in the oceans and predator-prey interactions which ultimately will need incorporating into future models of marine carbon flux dynamics.*"

Further, my concern about the size of the cells is still somehow valid. The problem is that the reader will see the electron micrographs in Fig. 1A and S1 where it is very clear that the cells differ in size and then they will be guided to the supplement where flow cytometry and fluorescence microscopy imply that there is no difference. I do not know why the electron micrographs give such a misleading picture, but this whole story is quite unsatisfactory. Especially, because the size and shape of cells would have such a strong influence on all the following experiments.

Authors: We completely acknowledge this comment and have been endeavouring to clarify this issue by re-imaging cultures by TEM over the last few months. Unfortunately, we have been having difficulties obtaining 'clean' images because it is difficult to get the wild type piliated cells to 'stick' to the TEM grids. We have been able to at least compare cell sizes between the wild type and pili mutant. This has shown that cell size is very similar to, and entirely consistent with,

our fluorescence microscopy and flow cytometry data. We have added these cell size measurements in the main text (line 226): “We further confirmed no apparent differences in cell size between the mutant and wild type strains ($2.05 \pm 0.42 \times 1.21 \pm 0.11 \mu\text{m}$ and $2.02 \pm 0.35 \times 1.26 \pm 0.13 \mu\text{m}$, respectively, consistent with published cell size data³¹), and observed no cell aggregates when grown in shaken or non-shaken liquid cultures (Fig S3 and S4).”

New version of supplementary Figure S1:

My concern regarding the bioinformatics analysis of a potential pilus machinery and figure 1E has only partly addressed. First, I have to apologize that I stated in my previous review that PilC is a monomeric subunit. This is not true, though several schemes show PilC as a single subunit. Based on structural data in heterotrophs this is a dimer and the authors should have noticed this. In the current figure 1E now this pilus part looks really strange, because PilC monomer is shifted to the left, leaving some open space on the right.

Authors: We are very grateful for this information. Our model of the pilus structure represented in Fig 1E was mainly based on Craig et al (2019 *Nat. Rev. Microbiol.*), but it is very similarly represented in other manuscripts (e.g. Taton et al 2020 *Nat. Commun.*; or Conradi et al 2020 *life*). As the reviewer points out, most publications may be mistaken as PilC seems to form a homodimer (i.e. Karuppiyah et. al. 2010 *Proteins*; Collins et. al. 2007 *J. Bacteriol.*; and Takhar et. al. 2013 *J. Biol. Chem.*). Hence, we have now represented PilC as a homodimer in Fig 1E.

New version of Figure 1:

Further PilN and PiO should be visualized as transmembrane proteins in the picture.

Authors: This has now been amended as suggested by the reviewer.

And I have serious doubts that PilM is missing. If you check the Synechococcus genome, there is a “standard” pilMNOQ (WH7803_2367-2364) operon. I agree pilM is not very well conserved, but if you do a hmmsearch with a model based on cyanobacterial PilM homologs you will detect a weak homology to the gene upstream of pilN. Homologs of WH7803_2367 are typically located directly upstream of pilNOQ genes as well. Obviously weak homology and synteny hint at the possibility that marine cyanobacteria evolved an atypical pilM. This should be discussed instead of writing that no pilM was detected.

Authors: This has now been removed in the main text (line 99), and changed in Fig 1E and discussed in the figure legend: “pilM and pilO were not identified by homology but the

SynWH7803_2367 and SynWH7803_2365 genes are suggested because they form a standard pilMNOQ operon as found in other species.”

I am also not so happy with the naming of the 3 PilT homologs. Again, there is a pilCTB operon in the genome and an additional copy of PilT containing a P-rich N-terminus. The pilT gene in the operon should be named pilT1, and the other copy pilT2 in accordance with the designation in other model organisms. The pilT (2) and pilT (3) naming scheme in the current paper is quite confusing.

Authors: Numbers have now been deleted in Fig 1E and the *pilCTB* operon is mentioned in the figure legend: “*While S. elongatus PCC 7942 encodes three pilT, only two were found in Synechococcus sp. WH7803, one being part of the characteristic pilCTB operon.*”

Further, the statement that the “six pilin like proteins may provide three distinct pili functions” is much too speculative in my opinion.

Authors: That this is speculation is now emphasised in the text (line 99): “*We speculate that the genetic cluster encoding the six pilin-like proteins (Fig 1C and 1D) may provide three distinct pili functions.*”

Minor:

In the legend to Fig. 5 you still say that it is a pilA1 mutant, which is not correct.

Authors: ‘PilA1 mutant’ has been changed to ‘pili mutant’.

Reviewer #3 (Remarks to the Author):

The authors have performed adequate experiments to solve most of my concerns. The images in Fig.S1B has not-so-clean background and I am not sure if those shown are pili or not, especially when you don't see a complete cell in it. Other than that I have no more questions.

Authors: We are very grateful to this reviewer for the helpful comments during the first review. We have now reformatted supplementary Fig S1B to make it clearer.

Reviewers' Comments:

Reviewer #1:

Remarks to the Author:

I do not have more comments, the authors adequately responded to my concerns. Just because the authors mentioned that they are not aware of alternative explanations: as you did not complement the mutant, it is possible that there is a second mutation in this strain which leads to the observed phenotype. You mention the paper by Nagar et al. which showed that quite a lot of mutations (including unknown genes) show a sedimentation phenotype. From other cyanobacteria it is also known that mutations in EPS-related genes show such a phenotype. Just to mention an alternative explanation....

Authors: We thank all reviewers and editor for their useful and constructive comments throughout the review process which have contributed to improve the quality and clarity of the manuscript.

REVIEWERS' COMMENTS

Reviewer #1 (Remarks to the Author):

I do not have more comments, the authors adequately responded to my concerns. Just because the authors mentioned that they are not aware of alternative explanations: as you did not complement the mutant, it is possible that there is a second mutation in this strain which leads to the observed phenotype. You mention the paper by Nagar et al. which showed that quite a lot of mutations (including unknown genes) show a sedimentation phenotype. From other cyanobacteria it is also known that mutations in EPS-related genes show such a phenotype. Just to mention an alternative explanation....

Authors: We agree that the complementation of the mutant would have been nice. Nevertheless, the lack of genetic systems in these environmentally-relevant and fastidious-to-grow microbes makes it impossible. We did, though, carry out the general recommendation of renown researchers in the field (e.g. Annegret Wilde or David Lea-Smith) which consists in testing various independent knockout mutants to rule out any possible second mutation which, by chance, would cause the observed phenotype. All three mutants obtained showed exactly the same phenotype. This is mentioned in the Methods section (line 273): *"All three transconjugant colonies that were picked had the same sinking phenotype, had doubly recombined (as checked by sequencing the overlapping regions) and were fully segregated."* Hence, the likelihood of the clear sinking phenotype to be caused by a second mutation is reduced enormously.

The reviewer also mentions other reasons for the sinking phenotype such as the production of EPS, (we believe) referring to its role in causing aggregation and sinking. First, marine picocyanobacteria lack most of the key genes necessary for EPS production (i.e. Pereira et al 2015 Scientific Reports, <https://doi.org/10.1038/srep14835>). And second, we prove by flow cytometry and microscopy (Supplementary Figures S3 and S4) that cells do not aggregate in these cultures.